



# Modelling Holocene peatland dynamics with an individual-based dynamic vegetation model

Nitin Chaudhary, Paul A. Miller and Benjamin Smith
Department of Physical Geography and Ecosystem Science, Lund University,
Sölvegatan 12, SE- 22362 Lund, Sweden

*Correspondence to:* N. Chaudhary (nitin.chaudhay@nateko.lu.se)

**Abstract.** Dynamic global vegetation models (DGVMs) are designed for the study of past, present and future vegetation patterns together with associated biogeochemical cycles and climate feedbacks. However, current DGVMs lack functionality for the representation of peatlands, an important store of carbon at high latitudes. We demonstrate a new implementation of peatland dynamics in a customised "Arctic" version of the dynamic vegetation model LPJ-GUESS, simulating the long-term evolution of selected northern peatland ecosystems and assessing the effect of changing climate on peatland carbon balance. Our approach employs a dynamic multi-layer soil with representation of freeze-thaw processes and litter inputs from a dynamically-varying mixture of the main peatland plant functional types; mosses, dwarf shrubs and graminoids. The model was calibrated and tested for a sub-arctic mire in Stordalen, Sweden, and validated at a temperate bog site in Mer Bleue, Canada. A regional evaluation of simulated carbon fluxes, hydrology and vegetation dynamics encompassed additional locations spread across Scandinavia. Simulated peat accumulation was found to be generally consistent with published data and the model was able to capture reported long-term vegetation dynamics, water table position and carbon fluxes. A series of sensitivity experiments were carried out to investigate the vulnerability of high latitude peatlands to climate change. We found that the Stordalen mire may be expected to sequester more carbon in the first half of the 21st century due to milder and wetter climate conditions, a longer growing season, and $CO_2$ fertilization effect, turning into a carbon source after mid-century because of higher decomposition rates in response to warming soils.

## 1 Introduction

Peatlands are a conspicuous feature of northern latitude landscapes (Yu et al., 2010), of key importance for regional and global carbon balance and potential responses to global change. In the past 5-10 thousand years they have sequestered approximately 200-550 Pg C across an area of approximately 3.5 million $km^2$ (Gorham, 1991; Turunen et al., 2002; Yu, 2012). Peatlands are also considered one of the major natural sources of methane, contributing significantly to the greenhouse effect (IPCC, 2013; Lai, 2009; Whiting and Chanton, 1993). The majority of northern peatland areas coincide with low altitude permafrost (Wania et al., 2009a). Permafrost changes the peat accumulation process by altering plant productivity and decomposition, affecting the carbon sequestration rate (Robinson and Moore, 2000). Thawing of permafrost exposes the organic carbon stored in the frozen soil which then becomes available for decomposition by soil microbes (Zimov et al., 2006).

Dynamic global vegetation models (DGVMs) are used to study past, present and future vegetation patterns from regional to





global scales, together with associated biogeochemical cycles and climate feedbacks, in particular through the carbon cycle (Friedlingstein et al., 2006; Sitch et al., 2008; Smith et al., 2001; Strandberg et al., 2014; Zhang et al., 2014). Only a few DGVMs include representations of the unique vegetation, biophysical and biogeochemical characteristics of peatland ecosystems (Kleinen et al., 2012; Tang et al., 2015; Wania et al., 2009a, b). Model formulations of peat accumulation and decay have been proposed and demonstrated at the site scale (Frolking et al., 2010) but have not yet, to our knowledge, been implemented within the framework of a DGVM, or applied at larger spatial scales than a single study site or landscape.

There is a scientific consensus that the climate is likely to warm in the coming century, and that the warming will be amplified in northern latitudes, relative to the global mean trend (IPCC, 2013). Current climate models predict that the northern high latitudes where most of the peatlands and permafrost areas are present could experience warming of more than 5°C by 2100 (Christensen et al., 2007; Hinzman et al., 2005; IPCC, 2013). The warm climate may alleviate the constraints on biological activity imposed by very low temperatures, leading to higher productivity and decomposition rates. The resultant shift in the balance between plant production and decomposition will alter the carbon balance, potentially leading to enhanced carbon sequestration in some peatlands (Charman et al., 2013; Yu, 2012) while inducing a carbon ($CO_2$ and $CH_4$) source in others (Fan et al., 2013; Ise et al., 2008; Wieder, 2001). Permafrost peatlands may respond quite differently to non-permafrost peatlands in changing climate conditions. Increases in soil temperature may accelerate permafrost decay (Åkerman and Johansson, 2008) and thereby modify the moisture balance of the peat soil, which could in turn alter the above ground vegetation composition and carbon balance of the permafrost peatlands (Christensen et al., 2004; Johansson et al., 2006).

We demonstrate a new implementation of peatland dynamics in the dynamic vegetation model LPJ-GUESS aiming to emulate the long-term dynamics of northern peatland ecosystems and to assess the effect of changing climate on peatland carbon balance at the regional scale and across climatic gradients. We build on previous work by implementing a dynamic multi-layer approach (Frolking et al., 2010; Hilbert et al., 2000) to peat formation and composition with existing representations of soil freezing-thawing functionality, plant physiology and peatland vegetation dynamics (Wania et al., 2009a) in a customised "Arctic" version of LPJ-GUESS (Miller and Smith, 2012). Uniquely among existing large-scale (regional-global) models, we thus account for feedbacks associated with hydrology, peat properties and vegetation dynamics, providing a basis for understanding how these feedbacks affect peat growth on the relevant centennial-millennial time-scales and in different climatic situations. We evaluate the model at a range of observational study sites across the northern high latitudes, and perform a model sensitivity analysis to explore the potential fate of peatland carbon in response to variations in temperature, atmospheric $CO_2$ and precipitation change in line with 21st century projections from climate models.

## 2 Model Overview

### 2.1 Ecosystem modelling platform



We employed a customised Arctic version of the Lund-Potsdam-Jena General Ecosystem Simulator (LPJ-GUESS; Smith et al., 2001; Miller and Smith, 2012) as the ecosystem modelling platform for our study. LPJ-GUESS is a process-based model that couples an individual-based vegetation dynamics scheme to biogeochemistry of terrestrial vegetation and soils (Smith et al., 2001). Vegetation structure and dynamics follow an individual- and patch-based representation in which plant population

demography and community structure evolve as an emergent outcome of competition for light, space and soil resources among simulated plant individuals, each belonging to one of a defined set of plant functional types (PFTs) with different functional and morphological characteristics (see below).

In this paper, we employ a customised Arctic implementation of LPJ-GUESS that incorporates differentiated representations of hydrological, biophysical and biogeochemical processes characteristic of upland and peatland ecosystems of the tundra

and taiga biomes, as well as plant functional types (PFTs) specific to Arctic ecosystems (Fig. 1) (McGuire et al., 2012; Miller and Smith, 2012). Five PFTs characteristic of peatlands – mosses (M), graminoids (Gr), deciduous and evergreen low shrubs (LSS and LSE) and deciduous high shrubs (HSS) – are included in the present study. These PFTs have different parameterizations of physiological processes, for instance relating to photosynthesis, leaf thickness, carbon allocation, phenology, and rooting depth. Full parameters sets for these PFTs are given in Miller and Smith (2012).

A one-dimensional soil column is represented for each patch (defined below), divided vertically into four distinct layers: a snow layer of variable thickness, a litter/peat layer of variable thickness, a mineral soil column with a fixed depth of 2 m (with further sublayers of thickness 0.1 m), and finally a "padding" column of 48 m depth (with thicker sublayers) allowing to simulate accurate arctic soil thermal dynamics (Wania et al., 2009a). The insulation effects of snow, phase changes in soil water, precipitation and snowmelt input and air temperature forcing are important determinants of daily soil temperature

dynamics at different depths.

New functionality was incorporated in LPJ-GUESS in this study in order to represent the dynamics of peat formation and aggradation based on vegetation litter inputs and decomposition processes. To this end, we adapted the dynamic multi-layer approach used in Frolking et al., 2010 and Hilbert et al., 2000 generalised for regional application. The new implementation is detailed in sections 2.1.1-2.1.7 below.

**2.1.1 Litterfall**

Peat accumulation is determined by the annual addition of new layers of litter at the top of the soil column. Litter is characterized as fresh, undecomposed plant material composed of dead plant debris such as wood, leaves and fine roots. Different PFTs accumulate carbon in the litter pool at different rates according to their productivity, mortality and leaf turnover properties. Litter is assumed to decompose at a rate dependent on the PFT and tissue type it originates from (Table

1). Based on the studies, woody litter mass from shrubs decomposes relatively slowly because it is made up of hard cellulose and lignin. Similarly, moss also decomposes at much slower rate due to its recalcitrant properties (Aerts et al., 1999; Moore




et al., 2007). Fresh litter debris decomposes through surface forcing until the last day of the year. The decomposed litter carbon is assumed to go directly to the atmosphere while any remaining litter mass is treated as a new individual peat layer from the first day of the following year, which then underlies the newly accumulating litter mass. This layer can be
composed of up to 17 carbon components (g C m$^{-2}$), namely leaf, root, stem and seeds from shrubs, mosses and graminoids (see Table 1) and the model keeps a track of these layer components as they decompose through time.

### 2.1.2 Peat accumulation and decomposition

Peat is partially decomposed litter mass. Accumulation occurs when net primary productivity (NPP) is higher than the decomposition rate, leading to carbon accumulation. Two functionally-distinct layers, the acrotelm and catotelm, are found
in most peatland sites. The acrotelm is the top layer in which water table fluctuates leading to both aerated and anoxic conditions. Due to uneven wetness, litter decomposes aerobically as well as anaerobically in the acrotelm (Clymo, 1991; Frolking et al., 2002). This layer also plays the critical role in determining plant composition. The catotelm exists below the permanent annual water table position (WTP) and remains waterlogged throughout the year, creating anoxic conditions, which in turn attenuate the decomposition rate and promote peat accumulation. The boundary between these two layers is
marked by the transition from the living plant parts to the dead plant parts and annual WTP.

Our model implicitly divides the total peat column into two parts—acrotelm and catotelm—demarcated by annual WTP, as determined by the hydrology scheme described below. Every year, a new litter layer is deposited over previously accumulated peat layers. After several years due to high carbon mineralization rates in the acrotelm layer (or upper peat layers above the annual WTP), the litter mass losses its structural integrity and transforms into peat, eventually becoming
integrated into the saturated rising catotelm mass. The rate of change of total peat mass is the total peat production minus total peat loss due to decomposition:

$$\frac{dM}{dt} = A - K M \qquad (1)$$

where M (kg C m$^{-2}$) is the total peat mass, A is the annual peat input (kg C m$^{-2}$ yr$^{-1}$), and K is the decomposition rate (yr$^{-1}$).

Total peat depth is derived from the dynamic bulk density values calculated for individual peat layers. The decomposition
process is simulated on a daily time step based on the decomposability of the constituent litter types in each layer and the soil physical and hydraulic properties of that layer. This difference in decomposability between litter types is represented by the initial decomposition rate ($k_o$ – see Eq. 2 and Table 1) (Aerts et al., 1999; Frolking et al., 2001). The initial decomposition rates are assumed to decline over time using a simplified first order reduction equation (Clymo et al., 1998; Frolking et al., 2001):

$$k_i = k_o \left( \frac{m_t}{m_o} \right) \qquad (2)$$





where $i$ refers to a litter component in a certain peat layer, $k_o$ is the initial decomposition rate, $m_o$ is the initial mass and $m_t$ is the mass remaining after some point in time (t). Peat water content and soil thermal dynamics are simulated at different depths (see below) and have a multiplicative effect on the daily decomposition rate (K) of each litter component in each layer following Lloyd and Taylor (1994) and Ise et al. (2008):

$$K_i = k_i T_m W_m \tag{3}$$

where $k_i$ is the decomposition rate of the layer i component (see Eq. 2) and $T_m$ and $W_m$ are the temperature and moisture multipliers, respectively. Following Ise et al. (2008), we assume that peat decomposition is highest at field capacity and lowest during very wet conditions. However, we allowed the peat to decompose in very dry conditions when the annual WTP drops below –400 mm (WTP takes negative (positive) values when the water table is below (above) the peat surface) and the volumetric water content ($\theta$) goes below 0.01 in the peat layers (Eq. 4 and Table 2).

$$W_m = \begin{cases} 1.0 - (1.0 - 0.025)\left(\frac{\theta - \theta_{opt}}{1.0 - \theta_{opt}}\right)^{\alpha}, & \theta > \theta_{opt} \\ 1.0 - \left(\frac{\theta_{opt} - \theta}{\theta_{opt}}\right)^{\alpha}, & \theta > 0.01 \text{ and } \theta \leq \theta_{opt} \\ \beta, & \theta \leq 0.01 \text{ and WTP} < -400 \end{cases} \tag{4}$$

where $\theta_{opt}$ is the field capacity and optimum volumetric water content when $W_m$ becomes 1.0 and $\alpha$ is a parameter that affects the shape of the dependency of decay on $\theta$, set to 5.0. The temperature multiplier is exponentially related to the peat temperature (see Eq. 5 and Table 2) (Frolking et al., 2002). Peat is assumed not to decompose under frozen conditions when the fraction of ice content is greater than zero.

$$T_m = \begin{cases} 0, & T_i < T_{min} \text{ and } I > 0 \\ \left(\frac{T_i - T_{min}}{|T_{min}|}\right)^{0.5}, & T_{min} < T_i < 0°C \\ Q_{10}^{T_i/10}, & T_i > 0°C \end{cases} \tag{5}$$

where $T_i$ is the peat temperature in peat layer (i), $T_{min}$ is the lowest temperature below which heterotrophic decomposition ceases, I is the ice content in each peat layer (i) and $Q_{10}$ is the proportional increase in decomposition rate for a 10°C increase in temperature.

Compaction and the loss of peat mass due to decomposition modify the structural integrity of peat layers (Clymo, 1984) potentially inducing changes in bulk density with depth. Some previous studies have found that the lower bulk density of newly accumulated peat layers increases as peat decomposes and becomes compressed due to overlying peat mass (Clymo, 1991) although bulk density often shows no net increase with depth in the catotelm. Following Frolking et al. (2010), we assume that bulk density is a non-linear function of total mass remaining ($\mu = M_t/M_o$) (see Eq. 6 and Table 2).





$$\rho(\mu_i) = \rho_{min} + \frac{\Delta\rho}{1+\exp\left(-(40(1-\mu_i)-34)\right)} \tag{6}$$

where $\rho_{min}$ is the minimum bulk density, $\Delta\rho$ is the difference between this minimum and a maximum bulk density, $\mu_i$ is the total mass remaining in peat layer $i$, $M_o$ is the initial peat layer mass and $M_t$ is the peat layer mass remaining after some point in time.

### 2.1.3 Permafrost/Freezing-thawing cycle

Freezing and thawing of peat and mineral soil layers is an important feature in permafrost peatlands, determining plant productivity, decomposition and hydrological dynamics (Christensen et al., 2004; Johansson et al., 2006; Wania et al., 2009b). To simulate permafrost, peat layer decomposition and cycles of freezing and thawing, the soil temperature at different depths must be calculated correctly. In the Arctic version of LPJ-GUESS as described by Miller and Smith (2012), mineral soil layers (i.e. below the peat layers added in this study) are subdivided into 20 sublayers of 10 cm thickness to calculate soil temperature at different depths. In our implementation, new peat layers are added on top of these mineral soil layers. To overcome computational constraints for millennial simulations we aggregate the properties of the individual annual peat layers into thicker sublayers for the peat temperature calculations, beginning with three sublayers of equal depth and adding a new sublayer to the top of previous sublayers after every 0.5 m of peat accumulation. The result is a soil column with a dynamic number of peat sublayers, 20 mineral soil layers and multiple "padding" layers to a depth of 48 m. A single layer of snow is included, as in existing versions of the model. Following Wania et al. (2009a), the soil temperature profile in each layer is calculated daily by numerically solving the heat diffusion equation. Soil temperature is driven by surface air temperature which acts as the upper boundary condition. Soil temperature in each annual peat layer is then updated daily and equal to the numerical sublayer to which it belongs. The amount of water and ice present in the sublayers together with their physical composition (mineral, organic or peat fractions) determine the thermal properties (soil thermal conductivities and heat capacities) of each sublayer. Freezing and thawing of soil water (see below) is modelled using the approach in following Wania et al. (2009a). The fraction of air and water is updated daily based on the soil temperature in each sublayer while the fraction of peat and organic matter is influenced by the degree of peat layer decomposability. In the sublayers, the fraction of mineral content is based on Hillel (1998). A full description of the soil temperature and permafrost scheme in the Arctic version of LPJ-GUESS is available in Miller and Smith (2012) and references therein.

### 2.1.4 Hydrology

Precipitation is the major source of water input in the majority of peatlands. In our model, precipitation is treated as rain or snow depending upon the daily surface air temperature. When temperature falls below the freezing point (0°C assumed), water is stored as a snow above the peat layers. Snow melts when the air temperature rises above the freezing point and is also influenced by the amount of precipitation on that day (Choudhury et al., 1998). We assume that the peatland can hold




water up to +20 cm above the peat surface. Water is removed from the peat layers through evapotranspiration, drainage, surface and base runoff. A traditional water bucket scheme is adopted to simulate peatland hydrology:

$$W = P - ET - R - DR \pm LF \tag{7}$$

where W is the total water input, P is the precipitation, ET is the evapotranspiration rate, R stands for the surface runoff, DR for the drainage and LF is the lateral flow within the landscape depending upon the relative position of the patch. WTP is updated daily based on existing WTP, W, the total drainage porosity and permeability of the peat layers. WTP is expressed in mm in this paper, with a value of 0 indicating a water table at the peat surface.

Evaporation can only occur when the snowpack is thinner than 1 cm and is calculated following the approach of Gerten et al. (2004), as in the standard version of LPJ-GUESS:

$$ET\,(WTP) = 1.32 \cdot E \cdot W_c^2 \cdot F \tag{8}$$

where E is the climate-dependent equilibrium evapotranspiration (mm), $W_c$ is the water content on the top 10 cm of the peat soil and F is the fraction of modelled area subject to evaporation.

Runoff is an exponential function of WTP (Wania et al., 2009a):

$$R = BR + \begin{cases} e^{0.01}\ WTP, & WTP > TH \\ 0, & WTP \leq TH \end{cases} \tag{9}$$

where TH is the WTP threshold (Table 2) and BR is the base runoff proportional to the total peat depth (D) and the base runoff is estimated as:

$$BR = u\,D \tag{10}$$

where u is a parameter (see Table 2) which determines rate of increase in the base runoff with increase in the peat depth (D) (Frolking et al., 2010). Loss of the water through drainage/percolation depends on the permeability of peat layers and the saturation limit of the mineral soil underneath. Percolation ceases if the mineral layers are saturated with water, incoming rainfall or snowmelt leading instead to an increase in WTP. We make the assumption that peat layers become highly compressed under accumulating peat mass and humified by anoxic decomposition (Clymo, 1991). This results in declining permeability, affecting the flow of water from the peat layers to the mineral soil. The permeability of each peat layer (i) is calculated as a function of peat layer bulk density (Eq. 11) (Frolking et al., 2010). The amount of water draining from the peat column to the mineral soil is calculated by integrating permeability across all the peat layers (i).

$$\kappa_i = 400\ e^{-0.075\rho_i} \tag{11}$$



where $\kappa_i$ is the permeability (0-1) and $\rho_i$ is the bulk density of peat layer (i). Change of porosity ($\Phi$) due to compaction is captured by a relationship to bulk density:

$$\Phi_i = 1 - \frac{\rho_i}{\rho_o} \tag{12}$$

where $\rho_o$ is the particle bulk density of the organic matter (see Table 2). Finally, water infiltrating from the peat to the mineral soil layers is treated as the input to the standard LPJ-GUESS hydrology scheme described in Smith et al. (2001) and Gerten et al. (2004).

### 2.1.5 Root distribution and water uptake

In the customized Arctic version of LPJ-GUESS, the mineral soil column is 2 m deep and partitioned into two layers, an upper mineral soil layer of 0.5 m and lower mineral soil layer of 1.5 m. The fraction of roots in these two layers is prescribed for different PFTs (Table 1) and used to calculate daily water uptake. Dynamic peat layers on top of the mineral soil layers necessitated a modification to the way plants access water from both the peat layers and the underlying mineral soil. In the beginning of the peat accumulation process, plant roots are present both in peat and upper and lower mineral soil layers but their mineral soil root distribution declines linearly as peat grows (see Fig. 2) and the corresponding mineral layer reduction is used to access water from the peat layers. Mosses are assumed only to take up water from the top 50 cm of the mineral soil and shallow peat surface in the beginning but once the peat depth exceeds 50 cm they only take water from the peat layers. Other PFTs can continue to take up water both from the mineral and peat soils until peat depth reaches 2 m, and from the peat soil thereafter.

### 2.1.6 Establishment and mortality

PFTs are able to establish within prescribed bioclimatic limits reflective of their distributional range (Miller and Smith, 2012) but are also limited by the position of the annual-average WTP. Shrubs are vulnerable to waterlogged and anoxic conditions (Malmer et al., 2005) and establish only when annual WTP deeper than -25 cm below the surface. Mosses and graminoids, by contrast, thrive in wet conditions and establish under WTP +5 to -50 cm (mosses) and above -10 cm (graminoids). The establishment function is implemented once per annual time step, based on mean WTP for the previous 12 months.

LPJ-GUESS includes a prognostic wildfire module (Smith et al., 2014; Thonicke et al., 2001). In high-latitude peatlands, the risk of natural fire events increases in prolonged dry and warm conditions and this is simulated by the model. Fires lead to vegetation mortality but are assumed not to lead to combustion of peat carbon in our implementation.

### 2.1.7 Microtopographical structure



Many studies have highlighted the importance of surface micro-formations in peatland dynamics (Belyea and Baird, 2006; Belyea and Malmer, 2004; Nungesser, 2003; Pouliot et al., 2011; Sullivan et al., 2008; Weltzin et al., 2001). The patterned surface creates a distinctive environment with contrasting plant cover, nutrient status, productivity and decomposition rates

in adjacent microsites. Such spatial heterogeneity is ignored in most peatland modelling studies, but can be critically important for peatland development and carbon balance. In our approach, multiple vegetation patches are simulated to account for such spatial heterogeneity. The model is initialised with a random surface represented by uneven heights of individual patches (10 in the simulations performed here). Water is redistributed from the higher elevated sites to low depressions through lateral flow (see Eq. 7). We equalize the WTP of individual patches according to the mean WTP of the

landscape. The higher patches loses water if the WTP is above the mean WTP of the landscape while the lower patches receive water. This in turn affects the PFT composition, productivity and decomposition rate in each patch, and peat accumulation over time.

## 2.2 Study area

### 2.2.1 Stordalen

The model was developed based on observations and measurements at Stordalen, a subarctic mire situated 9.5 km east of the Abisko Research Station in northern Sweden (68.36° N, 19.05° E, elevation 360 m a.s.l.) (Fig. 3). Stordalen is one of the

265 most studied mixed mire sites in the world and it has been part of the International Biological Program since 1970 (Rosswall et al., 1975; Sonesson, 1980). It is characterized by four major habitat types: (1) elevated, nutrient poor areas with hummocks and shallow depressions (ombrotrophic), (2) relatively nutrient rich wet depressions (minerotrophic), (3) pools and (4) small streams exchanging water from the catchment (Rosswall et al., 1975). Our simulations represent a mixed landscape of (1) and (2). The mire is mainly covered with mosses such as *Sphagnum fuscum* and *S. russowii*. Shrubs such as

*Betula nana*, *Andromeda polifolia* and *Vaccinium uliginosum* are present in dry hummock areas where the WTP remains relatively low, while hollows are mainly dominated by tall productive graminoids, e.g. *Carex rotundata* and *Eriophorum vaginatum* (Malmer et al., 2005). The Stordalen catchment is in the discontinuous permafrost zone. The elevated areas are mainly underlain with permafrost and wet depressions are largely permafrost free and waterlogged. Permafrost underlying elevated areas have been degraded as a result of climate warming in recent decades, with an increase in wet depressions

modifying the overall carbon sink capacity of the mire (Christensen et al., 2004; Johansson et al., 2006; Malmer et al., 2005). The annual average temperature of the Stordalen was -0.7°C for the period 1913-2003 (Christensen et al., 2004) and 0.49°C for the period 2002-2011 (Callaghan et al., 2013). The warmest month is July and coldest February. The mean annual average precipitation is low but increased from 304 mm (1961-1990) to 362 mm (1997-2007) (Johansson et al., 2013). Overviews of the ecology and biogeochemistry of Stordalen are provided by Sonesson (1980), Malmer et al. (2005) and

Johansson et al. (2006). Ecosystem respiration in Stordalen is lower than commonly observed in other northern peatlands due to low mean temperatures, a short frost-free season and the presence of discontinuous permafrost that keeps the thawed



soil cooler and restricts the decomposition rate (Lindroth et al., 2007). Based on radioisotope dating of peatland and lake sequences, Kokfelt et al. (2010) inferred that the peat initiation started ca. 4700 calendar years before present (cal. BP) in the northern part and ca. 6000 cal. BP in the southern part of the mire as a result of terrestrialisation.

**2.2.2 Mer Bleue**

To evaluate the generality of the model for regional (e.g. pan-Arctic) applications, we validated its performance against observations and measurements at Mer Bleue (45.40° N, 75.50° W, elevation 65 m a.s.l.), a raised temperate ombrotrophic bog located around 10 km east of Ottawa, Ontario (Fig. 3). The peat accumulation in this area initiated ca. 8400 cal. B.P and 290 the mean depth is around 4-5 m. The northwest arm of the bog is dome shaped with peat depths reaching 5-6 m near the central areas (Frolking et al., 2010; Roulet et al., 2007). The bog surface is characterized by hummock and hollow topography. This bog is mostly covered with Sphagnum mosses (*S. capillifolium, S. magellanicum*) and also dominated by a mixture of evergreen (*Chamaedaphne calyculata, Ledum groenlandicum, Kalmia angustifolia)* and deciduous shrubs (*Vaccinium myrtilloides*). A sparse cover of sedges (*Eriophorum vaginatum*) with some small trees (*Picea mariana, Larix* 295 *laricina, Betula populifolia*) is also present in the peatland (Bubier et al., 2006; Moore et al., 2002). The climate of the area is cool continental with the annual average temperature being 6.0±0.8°C for the period 1970 to 2000. The warmest month is July (20.9±1.1°C) and coldest January (-10.8±2.9°C). The average monthly temperature remains above 0°C from the April until November and above 10°C between May and September. The mean annual average precipitation is 910 mm of which 235 mm falls as a snow from December to March. The total precipitation is spread evenly across the year with a maximum 300 of 90 mm in July and a minimum of 58 mm in February.

**2.2.3 Additional evaluation sites**

To evaluate the performance of the model across high-latitude climatic gradients, simulations were performed at 8 locations across Scandinavia for which observations of peat depth and/or other variables of relevance to our study (ecosystem C fluxes, WTP, vegetation composition and cover) were available (Table 4). These sites represent different types of peatlands 305 with distinct initialization periods (from relatively new to old sites) and climate zones (from cold temperate to subarctic sites) (Fig. 3).

**2.3 Model forcing data**

The model requires daily climate fields of temperature, cloudiness and precipitation as input. Holocene climate forcing series for Stordalen and Mer Bleue were constructed by the delta-change method, i.e. applying relative anomalies derived from the 310 gridcell nearest to the location of the site from millennium time-slice experiments using the UK Hadley Centre's Unified Model (UM) (Miller et al., 2008), to the average observed monthly climate of the sites. Daily values were obtained by interpolating between monthly values for Stordalen from the year 5000 cal. BP and for Mer Bleue from the year 10000 cal. BP until the year 2000. For Stordalen we used the dataset of Yang et al. (2012) from the period 1913-1942, and for Mer



Bleue we used average monthly data from the CRU TS 3.0 global gridded climate data set (Mitchell and Jones, 2005) from
the period 1901 to 1930. We then linearly interpolated the values between the millennium time slices. This method conserves
the interannual variability for temperature and precipitation throughout the simulation. The version of the UM used in this
study was HadSM3, an atmospheric general circulation model (AGCM) coupled to a simple mixed layer ocean and sea ice
model with $2.5 \times 3.75°$ spatial resolution (Pope et al., 2000). The high spatial resolution (50 m), modern observed climate
dataset was developed by Yang et al. (2012) for the Stordalen site. In this dataset, the observations from the nearest weather
stations and local observations were included to take into account the effects of the Torneträsk lake close to the Stordalen
catchment. The monthly precipitation data (1913-2000) for Stordalen at 50 m resolution were downscaled from 10 min
resolution using CRU TS 1.2 data (Mitchell and Jones, 2005), a technique quite common for cold regions (Hanna et al.,
2005). The precipitation data was also corrected by including the influences of topography and also by using historical
measurements of precipitation from the Abisko research station record. Finally, monthly values of Holocene temperature
were interpolated to daily values, monthly precipitation totals were distributed randomly among the number (minimum 10)
of rainy days per month from the climate dataset and the monthly CRU values of cloudiness for the first 30 years from the
year 1901-1930 were repeated for the entire simulation period. We added random variability to the daily climate values by
drawing random values from a normal distribution with monthly mean (μ) and standard deviation (σ) of the monthly
observed climate were used for Stordalen from the period of 1913-1942 and for Mer Bleue, 30 years of monthly CRU values
from the period of 1901-1930 were utilized. For the additional evaluation sites, we used the randomly generated daily
climate CRU values of temperature and precipitation from the period 1901-1930. Past, annual atmospheric $CO_2$
concentration values from 5000 cal. BP for Stordalen and 10000 cal. BP for Mer Bleue to the year 2000 were obtained by
linear interpolation between the values used as a boundary conditions in the UM time-slice simulations (Miller et al., 2008).

The $CO_2$ concentration values used to force the UM simulations were linearly interpolated to an annually varying value
between prescribed averages for each millennium. From 1901 to 2000 observed annual $CO_2$ from atmospheric or ice core
measurements were used (McGuire et al., 2012).

### 2.4 Simulation Protocol

### 2.4.1 Holocene hindcast experiments

The model was first initialised for 500 years from "bare ground" using the first 30 years of Holocene climate data to attain an
approximate equilibrium of vegetation and carbon pools with respect to mid-Holocene climate. The mineral and peat layers
were forced to remain saturated for the entire initialization period. The peat decomposition, soil temperature and water
balance calculations were not started until the peat column became sufficiently thick (0.5 m). This initialisation strategy was
essential in order to avoid sudden collapse of the peat in very dry conditions. After initialization, the model was forced with
continuous Holocene climate from the year 4700 cal. BP until the year 1912, after which the observed climate of the
Stordalen site was used for the transient run until the year 2000. This experiment is referred as the standard model



experiment (STD). In the case of Mer Bleue, a similar procedure was adopted, but here the model was forced with continuous climate from the year 8400 cal. BP until the year 1900 and then the CRU climate was used for the transient run until the year 2000. Model parameters were identical in both cases, apart from those relating to local hydrology (u, TH – Eqs. 9 and 10) - see Table 2. This is to adjust the simulations with the local WTP. We refer to this experiment as the validation model experiment (VLD).

### 2.4.2 Hindcast experiment – regional climate gradient

The model was run at the eight additional evaluation sites spread across Scandinavia (Table 4; s2.2.3), comparing simulated peat accumulation to peat depth reported in the literature. Three sites were selected for additional evaluation; of carbon fluxes, WTP and dominant vegetation cover (Fig. 3 and Table 4 and 5). These simulations used a similar set up as in STD experiment with respect to bulk density and local hydrology.

Accurate prediction of total carbon accumulation across northern and high latitude peatlands is dependent on the right inception period, initial bulk density values and the local hydrology. The model was run within the most probable period of peat inception mentioned in the literature (Table 4).

### 2.4.3 Climate change experiment

To investigate the sensitivity of vegetation distribution, peat formation and peatland carbon balance to climate change, future experiments using RCP2.6 and RCP8.5 (Moss et al., 2010) 21st century climate change projections were performed, extending the STD experiment, which ends in 2000, until 2100. Climate output from the Coupled Model Intercomparison Project Phase 5 (CMIP5) runs with the MRI-CGCM3 general circulation model (GCM) was used to provide future climate forcing (Yukimoto et al., 2012). Climate sensitivity of MRI-CGCM3 is 2.60 K which is rather low compared to other models in CMIP5 (Andrews et al., 2012). Atmospheric CO2 concentrations for the RCP2.6 and RCP8.5 emissions scenarios were obtained from the website of the International Institute for Applied Systems Analysis (IIASA)- http://tntcat.iiasa.ac.at/RcpDb/. Simulations were performed for the Stordalen site. Responses of the model to single factor and combined future changes in temperature, precipitation and atmospheric $CO_2$ were examined in separate simulations (Table 3). Model output variables examined include cumulative peat age profile, total peat accumulation, net ecosystem exchange (NEE), annual WTP, active layer depth (ALD) and measures of vegetation PFT composition and productivity.

## 3 Results

### 3.1 Hindcast experiment

### 3.1.1 Stordalen

In the standard (STD) experiment, a total of 94.96 kg C m$^{-2}$ of peat was accumulated over 4700 years, leading to a cumulative peat depth profile of 2.11 m predicted for the present day (Fig. 4), comparable to the observed peat depth of 2.06



m reported by Kokfelt et al. (2010). The trajectory of peat accumulation since the mid-Holocene inception is also similar to the reconstruction based on radioisotope dating of the peat core sequence in combination with Bayesian modelling (Kokfelt et al., 2010) (Fig 4). Total NPP ranged from 0.06-0.18 kg C $m^{-2}$ $yr^{-1}$ during the simulation while the soil decay losses were between 0.05 and 0.15 kg C $m^{-2}$ $yr^{-1}$. Hence, the carbon uptake by the Stordalen mire ranged between -0.03 and 0.10 kg C $m^{-2}$ $yr^{-1}$ (Figs. 5a, 5c and A4). The long-term mean accumulation rate of the mire was 0.44 mm $yr^{-1}$ or 20 g C $m^{-2}$ $yr^{-1}$. Mean annual WTP drew down to -10 cm in the beginning and fluctuated between -10 to -25 cm for the entire simulation period, but decreased to a value below -25 cm in the last 100 years due to comparatively higher temperatures during this period (Fig. 5e). The model initially had an uneven surface where the majority of the patches were suitable for moss growth because of the shallow peat depth and an annual WTP near the surface (Figs. 5e and 6a). Moss-dominated areas accumulated more carbon as they become highly recalcitrant due to saturated conditions and low initial decomposition rate (see Table 1). At around 4300 cal. BP, shrubs started to establish because of a lower annual WTP as peat depth increased (Figs. 5e, 6a and A4). When the peat was shallow, plant roots were present in both the mineral and peat layers. Since the majority of lower peat and mineral layers were frozen, the water required for the plant growth was limited, which then limited the productivity of shrubs and graminoids. However, since the upper peat layers were not completely frozen the moss productivity was not limited to the same extent as they could take up the water from upper 50 cm of the peat surface (Figs. 6a and 7a). The total ice fraction was between 40 and 60% for the majority of the simulation period indicating that the peat soil was partially frozen from the beginning (Fig. A1). The fraction of ice present in the peat soil is influenced by mean annual air temperature (MAAT) and peat thickness (s2.1.3). Increasing MAAT can lead to a reduction in the fraction of ice present in the peatland if the peat is sufficiently shallow. However, in thicker peat profiles the influence of temperature was slower due to the thermal properties of the thicker peat layers. From Figure 7a, it is clear that at the end of the simulation period the lower layer (see X in Fig. 7a) was almost completely frozen but upper and middle layers were partially frozen (see Z in Fig. 7a) leading to a mean annual active layer depth (MAAD) of 0.64 m (Fig. 7c). When the peat layers had decomposed sufficiently and lost more than 70% of their original mass ($M_o$), their bulk density increased markedly. The mean annual bulk density of the full peat profile was initially around 40 kg $m^{-3}$, increasing to 50 kg $m^{-3}$ as the peat layers grew older and became highly decomposed after 4700 years, with the deepest layers often achieving bulk densities lower than 50 kg $m^{-3}$. The pore space and permeability are linked to the compaction of peat layers. Therefore, when the peat bulk density increased, pore space declined from 0.95 to 0.937 reducing the total permeability of peat layers that in turn reduced the amount of percolated water from the peat layers to the mineral soil.

### 3.1.2 Mer Bleue

In the VLD experiment, a total of 221.2 kg C $m^{-2}$ peat was accumulated over the simulation period, resulting in a peat profile of around 4.05 m (Fig. 4), which may be compared to the observed peat depth of 5 m reported by Frolking et al. (2010). The trajectory of peat accumulation is similar to the reconstruction based on radiocarbon dates for core MB930 by Frolking et al. (2010) for the first 6 kyr whereafter it diverges (Fig 4). The likely explanation for this late-Holocene divergence is discussed





in s4.1.1. Total NPP ranged from 0.1-0.5 kg C m$^{-2}$ yr$^{-1}$ in the course of the simulation while the soil carbon fluxes ranged

between 0.12 and 0.25 kg C m$^{-2}$ yr$^{-1}$. Therefore, the simulated carbon sequestration rate was in the range -0.2 to 0.3 kg C m$^{-2}$ yr$^{-1}$ (Figs. 5b, 5d and A4). NPP increased during the simulation period reaching 0.5 kg C m$^{-2}$ by the end of the simulation. Though both shrubs and mosses were the dominant PFTs from the beginning of the simulation, mosses were replaced by graminoids during the certain phases of peatland history and in the last 1000 years of the simulation (Fig. 6c). The mean accumulation rate was 0.48 mm yr$^{-1}$ or 26.3 g C m$^{-2}$ yr$^{-1}$. After the initialization period, annual WTP dropped to -50 cm and

later stabilised between -30 to -60 cm (Fig. 5f). The initial average bulk density of the peat profile was around 40 kg C m$^{-3}$, increasing to 93.8 kg C m$^{-3}$ as peat grew older while the pore space declined from 0.95 to 0.88.

### 3.2 Hindcast experiment – regional climate gradient

The majority of modelled peat depth values were in good agreement with published data (see Fig. 8 a, b and Table 4). At certain locations, notably Kontolanrahka (60.78° N, 22.78° E), Fajemyr (56.27° N, 13.55° E) and Lilla Backsjömyren

(62.41°N, 14.32°E) modelled peat depth was substantially different from observations reported in the literature (see Table 4 and Fig. 8). This could be because of the unavailability of site-specific climate forcing data (simulations were forced by interpolated station data from the CRU global gridded dataset), an incorrect initial bulk density profile or failure of the model to capture the local hydrological conditions. Fajemyr is a temperate tree bog and we have not considered litter coming from trees (T) and high evergreen shrubs (HSE) in this study, providing an additional potential reason for the underestimation of

simulated peat depth at this site. However, the modelled dominant vegetation cover, WTP and long-term apparent rate of carbon accumulation (LARCA)[1] were within the published ranges for all three sites with some discrepancies in short-term carbon fluxes (Table 5). Modelled dominant vegetation cover is similar to the observed cover except in Fajemyr where tree was also one of the dominant PFTs. Modelled LARCA values were also similar to observed values for the two sites (Fajemyr and Siikaneva) while no observed LARCA value was reported for Degerö Stormyr. Slightly wetter conditions were

simulated than observed at Degerö and Siikaneva. Modelled NEE was totally different from the observed fluxes for all the three sites.

### 3.3 Climate change experiments

In the future scenario experiments, the surface air temperature increased by approximately 4.8°C and 1.5°C in the T8.5 and T2.6 experiments by 2100, respectively, relative to the year 2000. The significantly higher temperature increase in the T8.5

experiment leads to complete disappearance of permafrost from the peat soil (Fig. 7c,d). Higher soil temperatures are associated with higher decomposition rates (Eq. 5) but since the MAAT is near to the freezing point (-0.7°C) at Stordalen a slight increase in temperature in the first 50 years leads only to a marginal increase in decomposition. However, melting of ice in the peat and mineral soils in combination with a milder climate and longer growing season lead to higher plant

---

[1] LARCA is calculated by dividing total cumulative carbon (peat thickness) by the corresponding time interval (basal age)





productivity (Fig. 6b and 7b). Therefore, the increase in decomposition is compensated by higher plant productivity leading

to an initial increase in the peat depth in the both T8.5 and T2.6 experiments (Fig. 9a and b). However, after 2050 decomposition dominates as temperature further increases leading to loss of substantial amount of carbon mass. Enhancement of plant photosynthesis due to $CO_2$ fertilization leads to increasing peat accumulation in both C8.5 and C2.6 experiments. Precipitation increases result in only a slight increase in peat depth in both the experiments (P8.5 and P2.6) because when the system is already saturated, any additional input of water will be removed at faster rates since evaporation

and surface runoff are positively correlated to WTP (see Eqs. 8 and 9, respectively). The combined effects of all drivers in FTPC8.5 and FTPC2.6 result in higher peat accumulation initially (see Fig. 9a and b), with reductions after 2050 as carbon mineralization rate increases as a result of higher temperature. The increase in carbon mineralization is also associated with thawing of permafrost. Before 2050 the fraction of ice is higher, restricting the decomposition rate. It is also evident from Fig. A2 that the vegetation and soil carbon fluxes are higher in both the experiments after 2050. In both the experiments

(FTPC8.5 and FTPC2.6), there is a loss of carbon after 2050 which stabilizes by the end of the century due to increased NPP (Fig. 9).

## 4 Discussion

### 4.1 Model performance

#### 4.1.1 Peat accumulation

Peat formation may be induced by a combination of several factors, among which climate, underlying topography, and local hydrological conditions are the important determinants (Clymo, 1992; Yu et al., 2009). In Stordalen, peat initiation started due to terrestrialisation of an open water area around ca. 4700 cal. BP in the northern part of the mire (Kokfelt et al., 2010)

while in Mer Bleue, the peatland formed ca. 8400 cal. BP (Frolking et al., 2010). We used these basal dates to start our model simulations. In the STD experiment, the simulated cumulative peat depth profile for the last 4700 years is consistent with the observed peat accumulation pattern (Kokfelt et al., 2010). The average increase in peat depth was simulated to be 2.11 m, which can be compared with the observed increase in peat depth of 2.06 m (Fig. 4). The simulated trajectory of the cumulative peat depth is also comparable to the observed data. In VLD experiment, the average increase in peat depth was

simulated to be 4 m, which can be compared to 5 m of observed peat depth (Frolking et al., 2010). The underestimation might be because the simulated annual productivity was slightly low, leading to relatively lower peat depth than observed. This discrepancy may also be traceable to the uncertainty in the climate model-generated palaeoclimate forcing of the peatland model. Studies of the influence of GCM-generated climate uncertainty (i.e. variations in climate output fields among GCMs) on carbon cycle model prediction, underline the high prediction error that can arise, for example in present-

470    day biospheric carbon pools and fluxes (Ahlström, 2016; Ahlström et al., 2013; Anav et al., 2013). Potential bias and errors in the predicted climate may be expected to be even higher in palaeoclimate simulations, not least due to the absence of instrumental observations for validating the models. Additional bias could arise due to the interpolation procedure used to



transform GCM output fields into monthly anomalies, required to force our model. These were generated by linearly interpolating between the climate model output, which is only available at 1000-year intervals. As such, the applied
anomalies do not capture decadal or centennial climate variability that can contribute to climate-forced variable peat accumulation rates on these timescales (Miller et al., 2008). Although the majority of the sites were in good agreement with the observed peat depth values in the regional gradient experiment, several factors may have contributed to poorer agreement for certain sites. In particular, a correct parameterization of local hydrological conditions, bulk density profile, climate forcing data and the right inception period are critical in determining the modelled long-term peat dynamics (Yu et al.,
2009), together with inclusion of suitable PFTs. Only the basal age was prescribed on a site-specific basis in our simulations (Table 4).

### 4.1.2 Coupled vegetation and carbon dynamics

Changes in vegetation cover significantly affect the long-term carbon fluxes due to differences in PFT productivity and
decay resistance properties of their litter (Malmer et al., 2005). In Stordalen, mosses and dwarf shrubs are the main peat forming plants present on hummocks and intermediate areas (Malmer and Wallen, 1996). Our results are largely in agreement with the observed changes in major PFTs during the last 4700 years of Stordalen history (Kokfelt et al., 2010). Mosses emerged as the dominant PFT at the beginning of the simulation, while 300-400 years after peat inception shrubs started establishing in the higher elevated patches as a result of a lowering of WTP (Figs. 5e and 6a). During the entire
simulation period, graminoids were not productive and deposited negligible amounts of litter mass. However, graminoids are known to have been present during certain phases of mire development so this aspect of the observed vegetation dynamics was not accurately captured (Kokfelt et al., 2010). One explanation can be the absence of decadal and centennial climate variability in our climate forcing datasets, resulting in "averaging out" of wet episodes needed for graminoids to be sufficiently competitive.

In Mer Bleue, mosses form the dominant vegetation cover together with low shrubs and graminoids. Though in general the model was able to capture these dynamics fairly well, we found some discrepancies in the beginning and at the end of the simulation. In the beginning, there were no graminoids while at the end the moss-dominated areas were replaced by graminoids due to submergence of lower patches, which is not reflected in the peat core analysis (Frolking et al., 2010).

For the additional evaluation sites, we found that the modelled dominant vegetation cover, LARCA and WTP were in good agreement with the observed values for the three selected sites at which this information was available. However, there were discrepancies between the observed and modelled values of short-term fluxes (Table 5). The variability in NEE is quite high and also very sensitive to local climate conditions, affecting prediction of these fluxes. However it is uncertain whether
recent annual observations of NEE necessarily reflect the long-term peatland carbon balance, in view of high variability on





multiple timescales. For example, observation of NEE during 1998-2001 in Mer Bleue averaged 70.2 g C m$^{-2}$ yr$^{-1}$ which dropped to 9.1 g C m$^{-2}$ yr$^{-1}$ in 2001-2002 due to dry summer (Lafleur et al., 2003 and Roulet et al., 2007).

Plant productivity simulated by our model in this study was generally quite low, as is generally observed in subarctic environments (Malmer et al., 2005). However, the NPP of mosses was comparatively higher than the dwarf shrubs because of two factors (Fig. 6a). The presence of permafrost (Fig. 7a) and an ALD near the surface (Fig. 7c) reduced the vascular plants' ability to take up water from the peat soil layers, reducing NPP and in turn affecting the total litter biomass (Fig. 5a). Mosses, however, could access water more easily because their uptake is largely above the ALD. The exposure to wind and snow drift may also contribute to reducing plant productivity (Johansson et al., 2006; Malmer et al., 2005) but these factors are not represented in the model. In the temperate conditions of Mer Bleue, plant productivity is quite high compared to subarctic conditions of Stordalen, as plant water uptake is not limited by permafrost conditions and it is also influenced by a longer growing season. In Mer Bleue, the total simulated NPP was low compared to that used as input to the modelling study by Frolking et al. (2010) but within the observed range reported by Moore et al. (2002). The lower simulated NPP in our model provides one explanation for relatively lower peat accumulation and peat depth, although agreement with the reconstructed peat accumulation trajectory is high for the first 6 kyr (Figs. 4 and 6c).

However, estimates of carbon fluxes derived from the flux tower measurements are not directly comparable with the long-term carbon fluxes derived from the peat core analyses (Belyea and Malmer, 2004; Silvola et al., 1996). LARCA for the two sites are 20 and 26.3 g C m$^{-2}$ yr$^{-1}$ respectively, which is near the reported mean for 795 peat cores from Finland (21 g C m$^{-2}$ yr$^{-1}$) (Clymo et al., 1998) and 127 accumulation records from northern peatlands (22.9 g C m$^{-2}$ yr$^{-1}$) (Loisel et al. 2014). The LARCA of all our evaluation sites also fall within reported ranges (see Table 4). Similarly, the mean annual simulated NEE (43.8 g C m$^{-2}$ yr$^{-1}$) for the last three decades (1971-2000) at the Stordalen site also falls within the recent observed range at the site of 8-45 g C m$^{-2}$ yr$^{-1}$ (Malmer et al., 2005; Malmer and Wallen, 1996). Christensen et al. (2012) found that the mean NEE of Stordalen mire during 2001-2008 was 46 g C m$^{-2}$ yr$^{-1}$ and for 2008-2009 it was 50 ± 17.0 g C m$^{-2}$ yr$^{-1}$ (Olefeldt et al., 2012; Yu, 2012). The mean NEE for 2001-2009 in our simulations was 50.7 g C m$^{-2}$ yr$^{-1}$, which is very near to the observed values. However, as discussed above, an exact comparison cannot yet be made as the carbon fluxes from the wet and semi-wet areas are not properly represented in our model, and the water borne fluxes are also not included in the calculation.

Water borne carbon fluxes (DOC) and CH$_4$ are not yet considered in our model (but are under development; e.g. Tang et al., 2015b) and inclusion of both would alter the NEE values we report above and in Figs. 5c and 5d. Both release and uptake components of NEE are relatively low in Stordalen compared to other peatlands (Nilsson et al., 2008; Olefeldt et al., 2012). The low ecosystem respiration is associated with low autotrophic respiration (Olefeldt et al., 2012) and the presence of permafrost which keeps the thawed peat soil cool and reduces the decomposition rate in the shallow thawed soil.



Temperature increase since the 1970's at Stordalen (Christensen et al., 2012) has caused the permafrost in the peat soil to thaw, leading to a predominance of wet sites dominated by graminoids in parts of the mire, affecting its overall vegetation composition and carbon fluxes (Christensen et al., 2004; Johansson et al., 2006). This situation was not captured by our simulation, where there is no such increase in graminoids (Fig. 6b). The increase in wet areas at Stordalen is however associated with peat soil subsidence during permafrost thaw and the resultant change in hydrological networks across the mire landscape (Åkerman & Johansson 2008), a complex physical process not included in our model. Another factor that contributed to the recent dynamics of the site is the influence of the underlying topography on the sub-surface flow and the addition of water through run-on from the surrounding catchment (Tang et al., 2015). Though we incorporated lateral exchange of water between the simulated patches, we ignored the effect of underlying topography that affects the water movement. In Stordalen, the southern and western parts of the mire are normally fed from higher areas centrally and to the east (Johansson et al., 2006), and recent warming has resulted in the runoff rate increasing from the elevated sites to the low lying areas that have slowly become increasingly waterlogged. Tang et al. (2015) showed the importance of including the slope and drainage area in order to distribute water within the catchment area, and demonstrated how these factors influence vegetation distribution and carbon fluxes in LPJ-GUESS.

## 4.2 Impact of climate change

### 4.2.1 Coupled vegetation and carbon dynamics

Some peatlands may sequester more carbon under warming climate conditions (Charman et al., 2013) while some may turn into carbon sources and degrade (Fan et al., 2013; Ise et al., 2008). For Stordalen, our simulations suggested that the temperature (T8.5 and T2.6) is the main factor which accelerates the decomposition in the peat soil after the year 2050. However, the rate of decomposition remains stable in the first half of the 21$^{st}$ century due to the presence of permafrost. The rise in atmospheric $CO_2$ concentration (C8.5 and C2.6) accelerates the plant productivity. An increase in precipitation (P8.5 and P2.6) has a very limited effect on peat growth as the mire has already been saturated and any additional input of water will be removed at a faster rate because the surface runoff and evaporation are positively correlated with WTP. The warmer and wetter future conditions, in combination with $CO_2$ fertilization (FTPC8.5 and FTPC2.6), would lead to increased moss productivity and a slight increase in shrub abundance (Figs. 6b and 9). The latter trend is consistent with widespread reports of expansion of tall shrubs in the second half of the 21$^{st}$ century in many parts of the Arctic and beyond (Loranty and Goetz, 2012; Sturm et al., 2005). Higher temperatures will result in earlier snowmelt and a longer growing season (Euskirchen et al., 2006), promoting plant productivity. Our results for both a strong warming (RCP8.5) and low warming (RCP2.6) scenario indicate that the limited increase in decomposition due to soil warming will be more than compensated by the increase in NPP in the first half of the 21st century, resulting in accelerated peat accumulation, but that the increase in decomposition outpaces the increase in NPP by around 2040, resulting in the loss of a substantial amount of carbon by the end of the 21$^{st}$ century (Fig 9).



### 4.2.2 Permafrost and climate warming

Temperature and precipitation are expected to increase at Stordalen in the coming decades (Saelthun and Barkved, 2003) and
alongside an increase in snow depth are expected to result in rapid rates of permafrost degradation and a thicker active layer
(Christensen et al., 2004; Johansson et al., 2013). Due to recent warming the ALD has already increased at Stordalen mire
and surrounding sites over the past three decades (Åkerman and Johansson, 2008). This event has also changed the surface
hydrology of the mire and in turn the vegetation distribution within the basin. ALD has increased between 0.7 and 1.3 cm per
year in different parts of the mire, accelerating to an average of around 2 cm yr$^{-1}$ in recent decades. In our results, we found
that simulated MAAD was around 0.67 m for 1972-2005, consistent with the observed MAAD of 0.58 m for the same period
(Christensen et al., 2004; Johansson et al., 2006). However, it should be noted that our model does not account for the large
observed impact of local variation in permafrost thaw on hydrological network and variability in wetness across the mire
landscape. According to Fronzek et al. (2006), a slight increase (1°C) in temperature and precipitation (10% increase) could
lead to widespread disappearance of permafrost throughout Scandinavia in the future. In one scenario, they found a complete
disappearance of permafrost by the end of the 21$^{st}$ century. Our results for Stordalen are consistent with this scenario: in the
FTPC8.5 experiment, permafrost completely disappears by 2050 due to climate warming (Figs. 7b and d). In the more
moderate warming of the FTPC2.6 experiment, permafrost thaws but does not disappear after the year 2050, leading to the
simulated MAAD of 1.75 m by 2100 (Fig. 7d).

### 5 Conclusion

Our results demonstrate that the incorporation of peatland and permafrost functionality in LPJ-GUESS provides a suitable
framework for assessing the combined and interactive responses of peatland vegetation, hydrology and soils to changing
drivers under a range of high latitude climates. Modelled peat accumulation, vegetation composition, water table position,
and carbon fluxes were found to be broadly consistent with published data for simulated localities in a range of high-latitude
climates. Climate change sensitivity simulations for the Stordalen mire suggest that peat will continue to accumulate in the
coming decades, culminating in mid-century (the year 2050), thereafter switching to a $CO_2$ source as a result of accelerating
decomposition in warming peatland soil. As a complement to empirical studies, our modelling approach can provide an
improved understanding of the long-term dynamics of northern peatland ecosystems at the regional scale, including the fate
of peatland carbon stocks under future climate and atmospheric change. In ongoing work, the model is being extended to
incorporate methane biogeochemistry and nutrient dynamics, and will be used to assess impacts of projected future changes
in climate and atmospheric $CO_2$ on peatland vegetation and greenhouse gas exchange across the Arctic. Coupled to the
atmospheric component of a regional Arctic system model, it is being used to examine the potential for peatland-mediated
biogeochemical and biogeophysical feedbacks processes to amplify or dampen climate change in the Arctic and globally.



**Acknowledgements**

This study was funded by the Nordic Top Research Initiative DEFROST and contributes to the strategic research areas Modelling the Regional and Global Earth System (MERGE) and Biodiversity and Ecosystem Services in a Changing Climate (BECC). We also acknowledge support from the Lund University Centre for the study of Climate and Carbon Cycle (LUCCI). We are also thankful to Anders Ahlström for providing the RCP dataset and Ulla Kokfelt for sharing age-depth

data of Stordalen mire.



**Figures:**

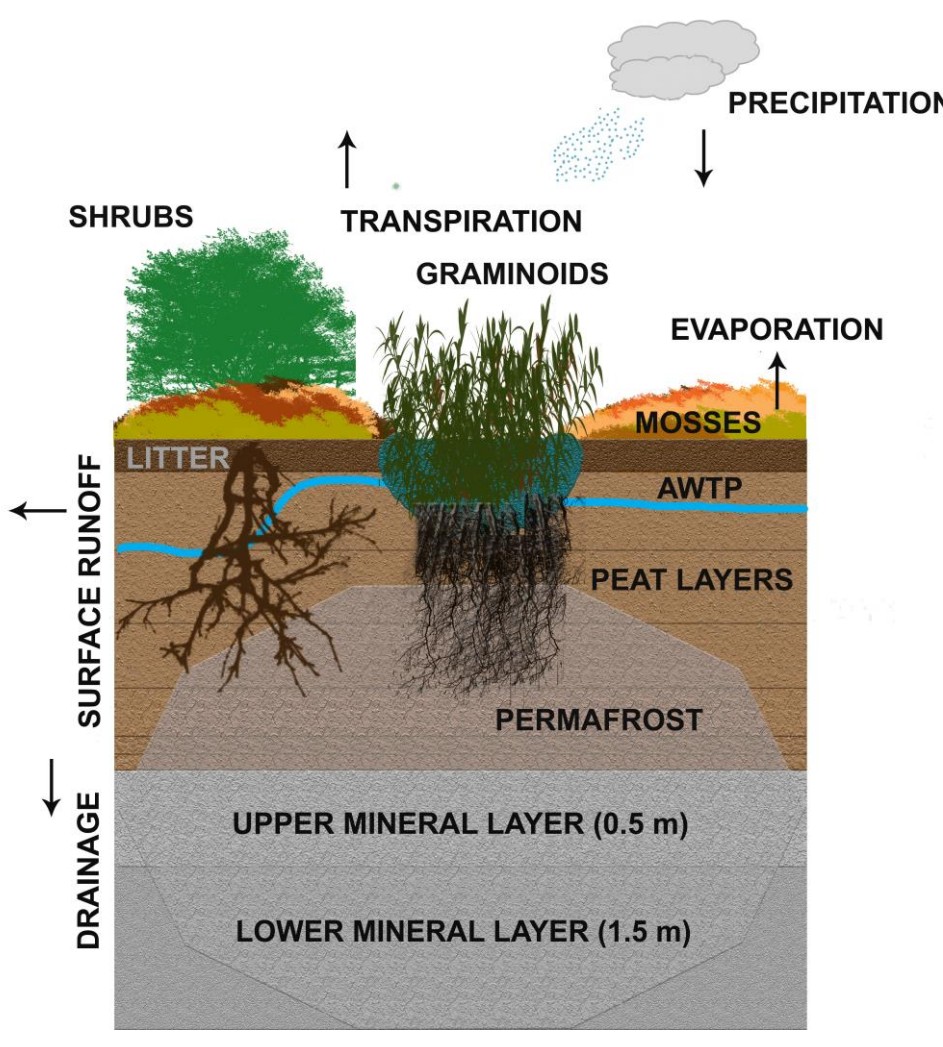

**Fig. 1.** Schematic representation of peatland structure and function in the implementation described in this paper. Dynamic peat layers deposit above the static mineral soil layers (0.5+1.5 m). In the shallow peat, plant roots are present in both mineral and peat layers. Once the peat becomes sufficiently thick (2 m), all roots are confined to the peat layers.



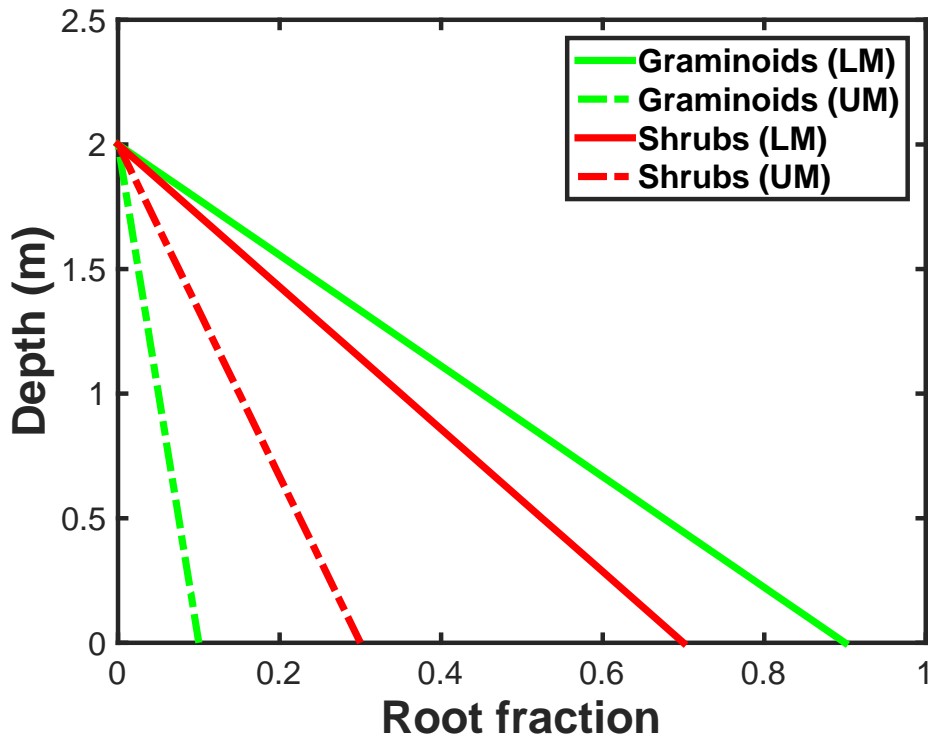

**Fig. 2.** Root fractions in the upper (UM) and lower mineral (LM) soil layers as a function of peat depth (m). The broken lines represent root fractions in UM soils and solid lines indicate fractions in the LM soil.



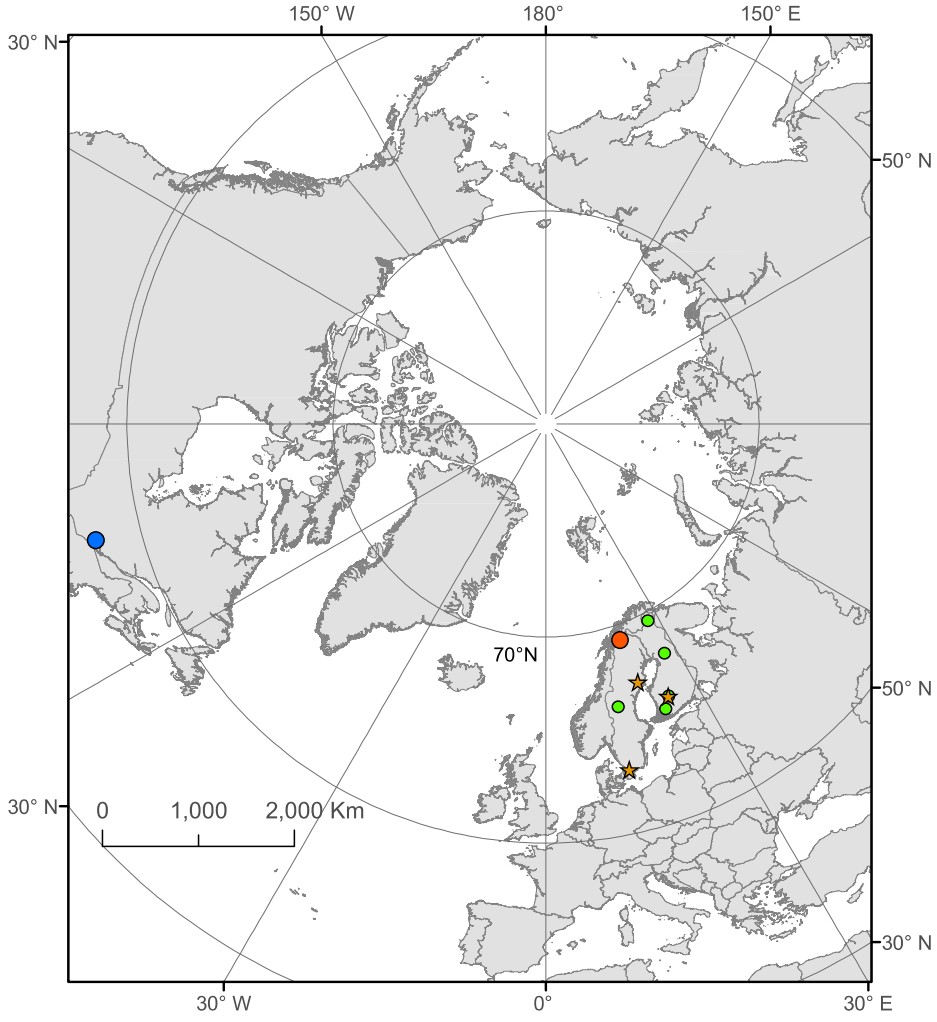

**Fig. 3.** Map showing the location of the evaluation site (in red), the validation site (in dark blue) and the distribution of regional gradient points across northern European (in green) used for validating the peat depth. Orange stars show the location of the three points used for the evaluation of peat depth, carbon fluxes, WTP and dominant vegetation cover.



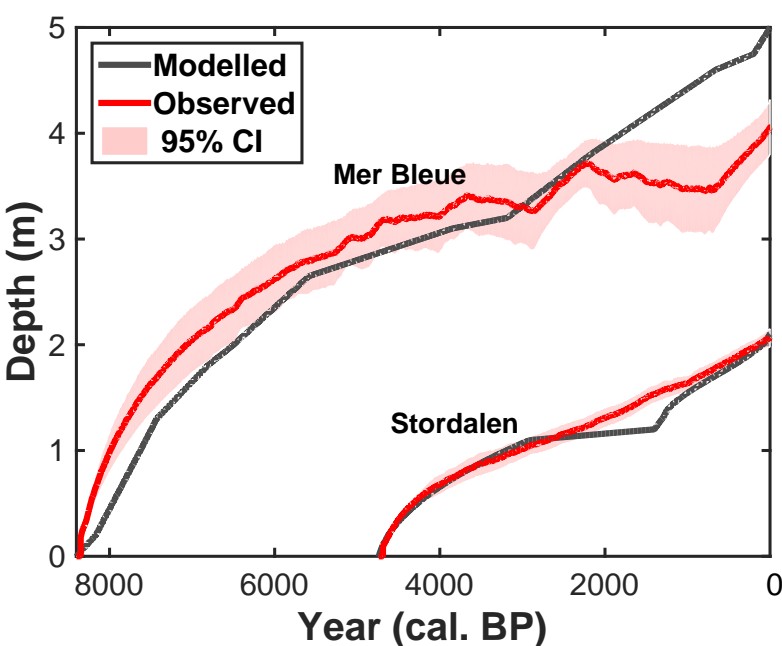

**Fig. 4.** Comparison of mean landscape simulated peat depth (m) with inferred ages of peat layers of different depths in peat cores from the Stordalen and Mer Bleue sites. The light red shaded area shows the 95% confidence interval (CI) inferred from the simulation data.





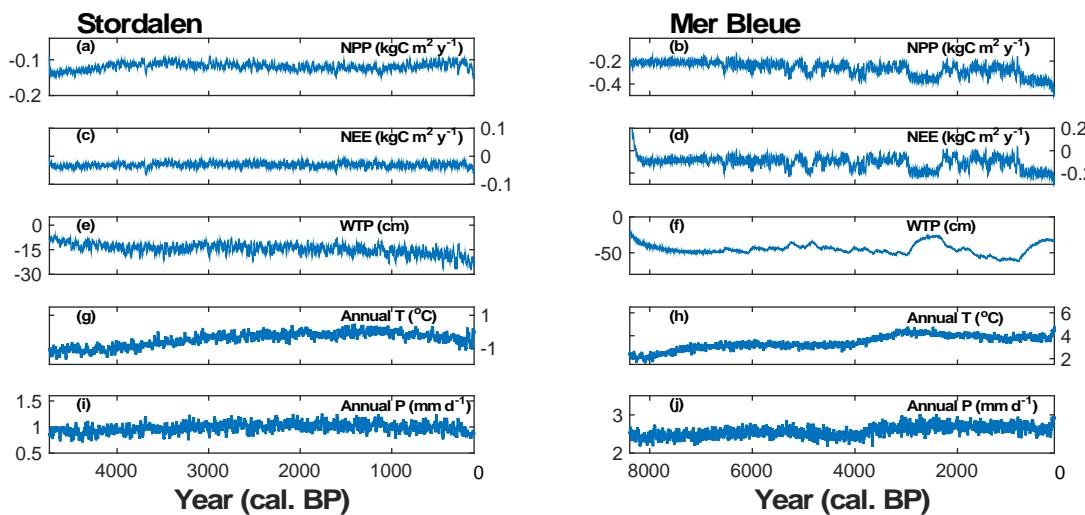

**Fig. 5.** Simulated annual average values (10-year moving average) of **(a, b)** net primary productivity (NPP), **(c, d)** net ecosystem exchange (NEE), **(e, f)** water table position (WTP), **(g, h)** temperature and **(i, j)** precipitation for the last 4700 years at the Stordalen mire and for the last 8400 years at Mer Bleue, respectively.





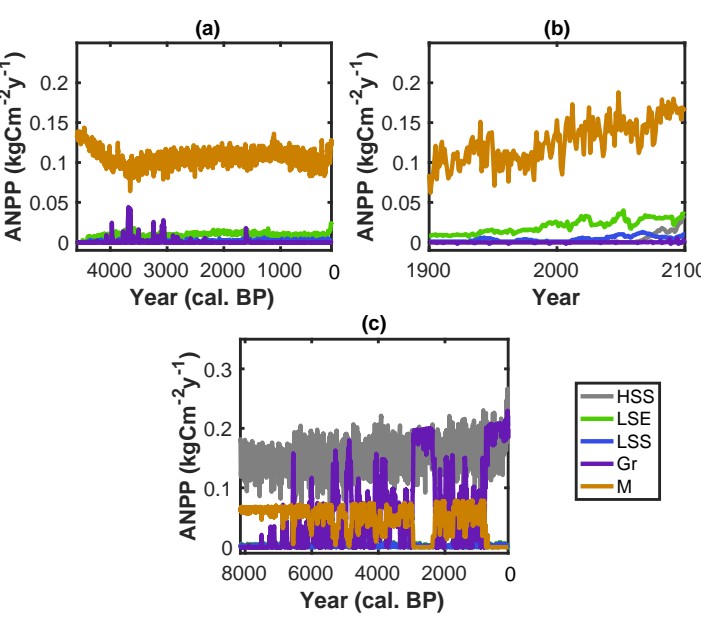

**Fig. 6.** Simulated annual net primary productivity (ANPP) (10-year moving average) of simulated PFTs **(a)** for the last 4700 years at the Stordalen site, **(b)** from the year 1900 to 2100 at the Stordalen site following RCP8.5 scenario (see Fig. A3 for RCP2.6 scenario) and **(c)** for the last 8400 years at the Mer Bleue site. Here HSS is high deciduous shrubs, LSE is low evergreen shrubs and LSS is low deciduous shrubs, Gr means graminoids and M is moss.





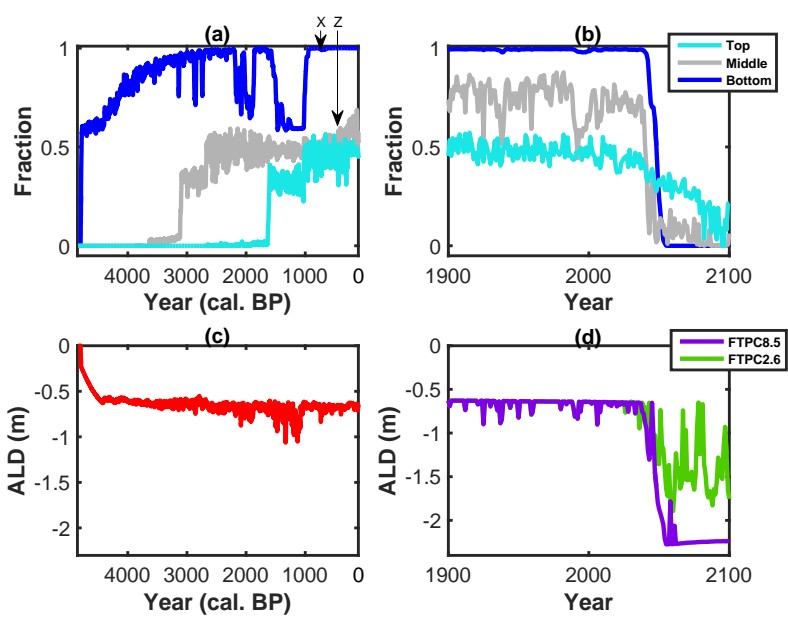

**Fig. 7.** (**a**) Total simulated ice fraction (10-year moving average) in the peat sublayers over 4700 years at Stordalen. The bottom sublayer was present from the beginning of the simulation while the middle sublayer formed once the peat depth reached to 1 m around 3200 cal. BP and the top sublayer was formed around 1200 cal. B.P. (**b**) Total simulated ice fraction from the year 1900 to 2100 using RCP8.5 scenario (see Fig. A3 for RCP2.6 scenario), (**c**) Total simulated mean active layer depth in the peat soil in September for the last 4700 years and (**d**) from the year 1900 to 2100 at Strodalen site using RCP8.5 scenario (FTPC8.5) and RCP2.6 scenario (FTPC2.6).




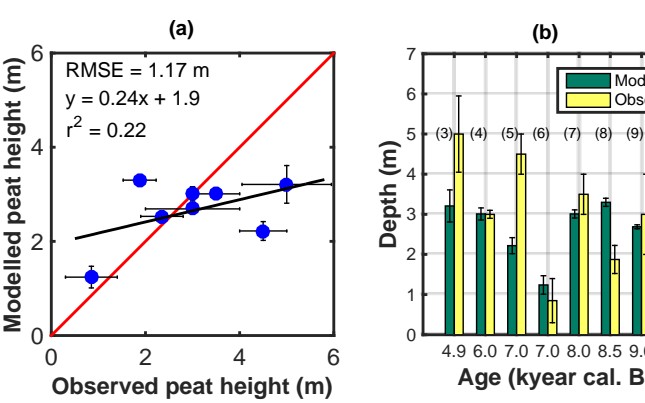

**Fig. 8. (a)** Scatter plot with range bars and **(b)** bar graph showing the comparison between modelled
and observed peat depth (m) with reported range bars (in black with yellow bars) at 8 locations across
Scandinavia. Corresponding site no. above the bars is described in Table 4.





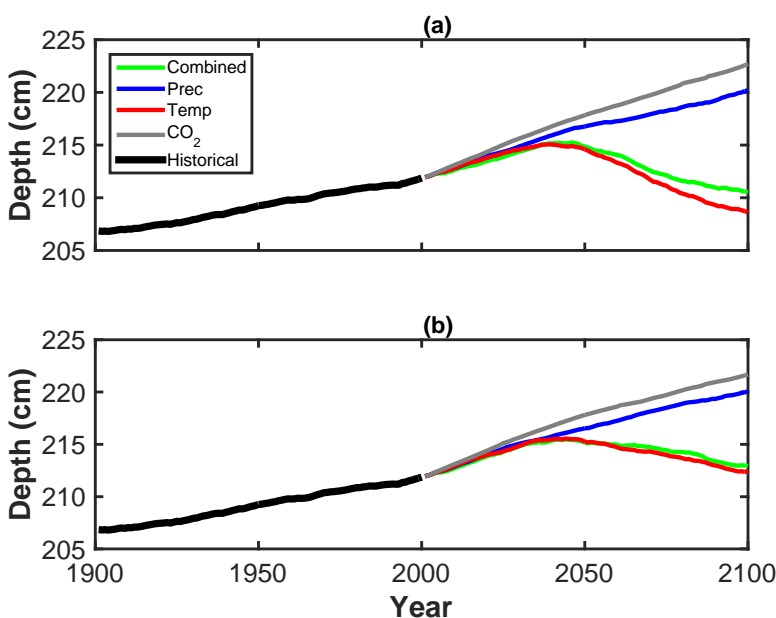

**Fig. 9.** Simulated peat depth (cm) in the future experiments using (a) RCP8.5 and (b) RCP2.6 forcing scenarios at Stordalen mire





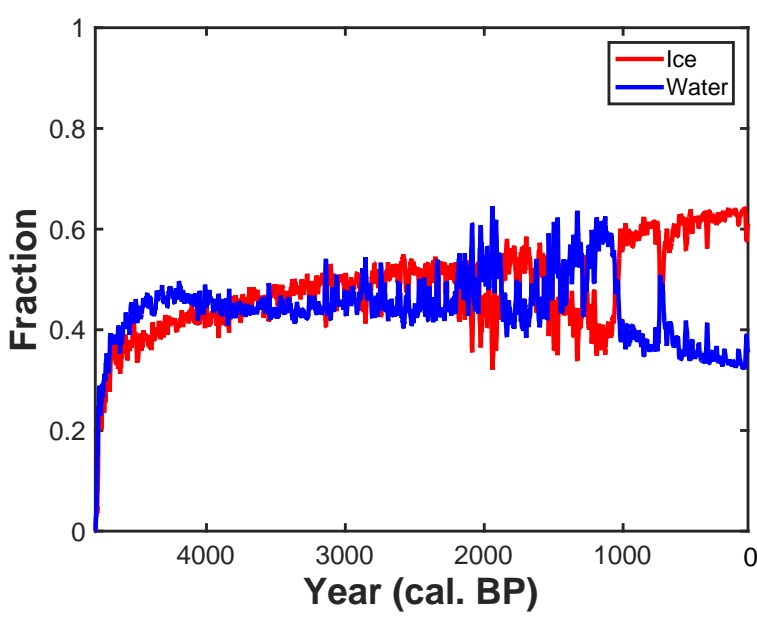

**Fig. A1.** Total simulated (10-year moving average) ice and water content in the peat soil for the last
4700 years at Stordalen mire





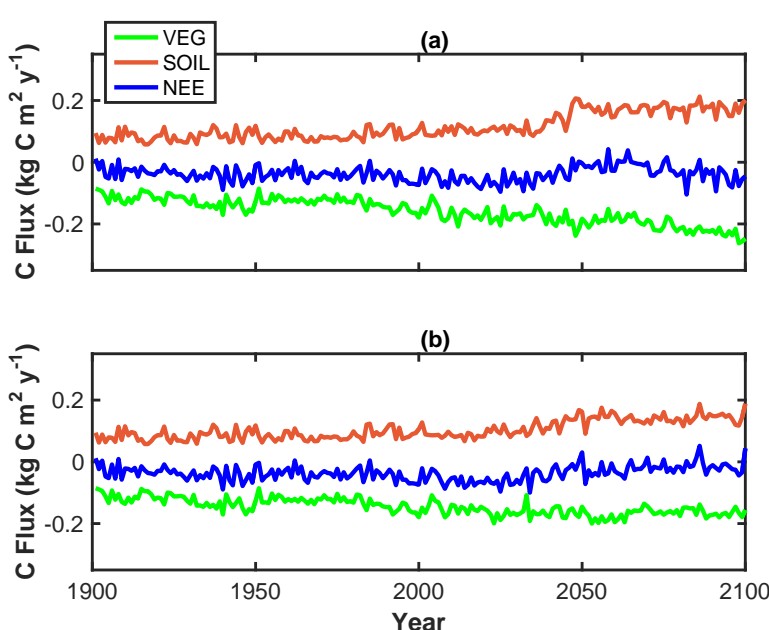

**Fig. A2.** Total simulated carbon fluxes from the vegetation (VEG) and soil (SOIL) and net ecosystem exchange (NEE) from the year 1900 to 2100 using **(a)** RCP8.5 (FTPC8.5) and **(b)** RCP2.6 (FTPC2.6) forcing scenarios at Stordalen mire





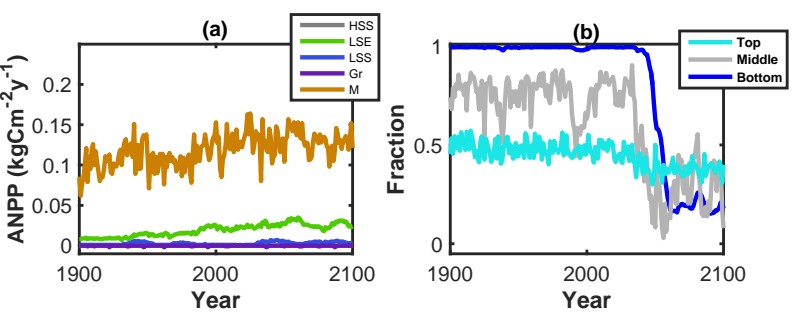

**Fig. A3.** Simulated annual net primary productivity (ANPP) and **(b)** Total simulated ice fraction in the
peat sublayers from the year 1900 to 2100 at the Stordalen site using RCP2.6 scenario (FTPC2.6)





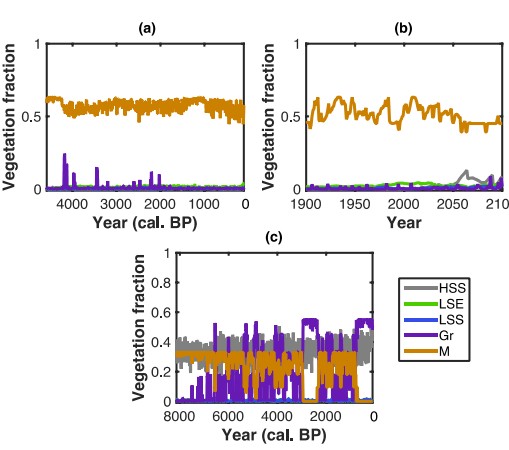

**Fig. A4.** Simulated vegetation fraction (10-year moving average) of simulated PFTs **(a)** for the last 4700 years at the Stordalen site, **(b)** from the year 1900 to 2100 at the Stordalen site following the RCP8.5 scenario and **(c)** for the last 8400 years at the Mer Bleue site. Here HSS is high deciduous shrubs, LSE is low evergreen shrubs and LSS is low deciduous shrubs, Gr denotes graminoids and M is moss.



**Table 1.** Plant functional types (PFTs) simulated in this study, showing representative example taxa, water table position (WTP) threshold for establishment and initial
decomposition rate for different litter fractions.

| PFT (abbreviation) | Representative taxa | WTP threshold (in mm) | Root fraction | | Litter fraction | Initial decomposition rate ($k_o$)[b] (yr$^{-1}$) |
|---|---|---|---|---|---|---|
| | | | Upper mineral soil (UM) | Lower mineral soil (LM) | | |
| High summergreen shrub (HSS) | *Salix spp., Betula nana* | <-250 | 0.7 | 0.3 | Wood | 0.055 |
| | | | | | Leaf | 0.1 |
| | | | | | Root | 0.1 |
| | | | | | Seed | 0.1 |
| Low evergreen shrub (LSE) | *Vaccinium vitis-idaea, Andromeda polifolia* | <-250 | 0.7 | 0.3 | Wood | 0.055 |
| | | | | | Leaf | 0.1 |
| | | | | | Root | 0.1 |
| | | | | | Seed | 0.1 |
| Low summergreen shrub (LSS) | *Vaccinuim myrtillus, Vaccinium uliginosum, Betula nana* | <-250 | 0.7 | 0.3 | Wood | 0.055 |
| | | | | | Leaf | 0.1 |
| | | | | | Root | 0.1 |
| | | | | | Seed | 0.1 |
| Graminoid (Gr) | *Carex rotundata Wg., Eriophorum vaginatum* | >-100 | 0.9 | 0.1 | Leaf | 0.1 |
| | | | | | Root | 0.1 |
| | | | | | Seed | 0.1 |
| Moss (M) | *Sphagnum* spp. | <-100 and >-500 | - | - | Leaf | 0.055 |
| | | | | | Seed | 0.055 |

---

[b] Aerts et al. (1999), Frolking et al. (2002) and Moore et al. (2007)





**Table 2.** Model parameter values used in standard (STD) and validation (VLD) model experiments

| Sl. no. | Parameter | Value | | Unit | Equation |
|---|---|---|---|---|---|
| | | STD | VLD | | |
| 1. | $\alpha$ | 5.0 | | - | Eq. (4) |
| 2. | $\beta$ | 0.064 | | - | Eq. (4) |
| 3. | $\theta_{opt}$ | 0.75 | | - | Eq. (4) |
| 4. | Tmin | -4 | | °C | Eq. (5) |
| 5. | $Q_{10}$ | 2 | | - | Eq. (5) |
| 6. | $\rho$min | 40 | | kg m$^{-3}$ | Eq. (6) |
| 7. | $\Delta\rho$ | 80 | | kg m$^{-3}$ | Eq. (6) |
| 8. | TH | -300 | -400 | mm | Eq. (9) |
| 9. | u | 0.45 | 0.0 | - | Eq. (10) |
| 10. | $\rho$o | 800 | | - | Eq. (12) |



**Table 3.** Summary of hindcast and global change experiments

| Experiment no. | Experiment name | Description of hindcast and future experiments from 2000 to 2100 |
|---|---|---|
| 1. | STD | Standard model experiment |
| 2. | VLD | Validation model experiment |
| 3. | T8.5 | RCP8.5 temperature only |
| 4. | P8.5 | RCP8.5 precipitation only |
| 5. | C8.5 | RCP8.5 $CO_2$ only |
| 6. | FTPC8.5 | RCP8.5 including all treatments |
| 7. | T2.6 | RCP2.6 temperature only |
| 8. | P2.6 | RCP2.6 precipitation only |
| 9. | C2.6 | RCP2.6 $CO_2$ only |
| 10. | FTPC2.6 | RCP2.6 including all treatments |



**Table 4.** Observed peat depth (m) compared with modelled peat depth (m), basal age, climatology, long-term apparent rate of carbon accumulation (LARCA) and total
accumulated carbon (kg C m$^{-2}$) for the calibrated and validation sites together with 8 grid points in the Scandinavian region

| | | | | | | | | | Modelled | | Observed | |
|---|---|---|---|---|---|---|---|---|---|---|---|---|
| Site no. | Site name | Peatland type | Country | Lat. (°N) | Lon. (°E) | MAAT (°C) | MAP (mm yr$^{-1}$) | Basal age (kyear cal. BP) | Total carbon (LARCA) kg C m$^{-2}$ (kg C m$^{-2}$ yr$^{-1}$) | Total peat depth range (average) (in meters) | Total peat depth range (average) (in meters) | Reference |
| 1. | Stordalen | Plasa mire | Sweden | 68.5 | 19.0 | -0.7 | 300 | 4.7 | 94.9 (20.0) | 1.9 - 2.2 (2.1) | 1.9 - 2.3 (2.1) | Kokfelt et al. (2010) |
| 2. | Mer Bleue | Temperate bog | Canada | 45.4 | -75.5 | 5.8 | 910 | 8.4 | 221.2 (26.3) | 3.6 - 4.4 (4.05) | 4.0 - 5.9 (4.9) | Frolking et al. (2010) |
| 3. | Kontolanrahka | Bog | Finland | 60.78 | 22.78 | 4.6 | 574 | 4.9 | 159.7 (32.5) | 2.7 - 3.4 (3.2) | 4.0 - 6.0 (5.0) | Valiranta et al. (2007) |
| 4. | Lakkasuo | Bog | Finland | 61.78 | 24.30 | 3.1 | 700 | 6.0 | 162.0 (27.0) | 2.9 - 3.2 (3.0) | 2.9 - 3.1 (3.0) | Tuittila et al. (2007) |
| 5. | Fajemyr | Temperate tree bog | Sweden | 56.27 | 13.55 | 6.2 | 700 | 7.0 | 128.2 (18.3) | 2.0 - 2.4 (2.2) | 4.0 - 5.0 (4.5) | Lund et al. (2007) |
| 6. | Kaamanen | Subarctic poor fen | Finland | 69.14 | 27.30 | -1.1 | 470 | 7.0 | 75.3 (10.8) | 1.1 - 1.5 (1.2) | 0.3 - 1.4 (0.9) | Aurela et al. (2004) |
| 7. | Degerö Stormyr | Boreal poor | Sweden | 64.18 | 19.55 | 1.2 | 523 | 8.0 | 166.0 (20.7) | 2.9 - 3.1 | 3.0 - 4.0 | Sagerfors et al. |





| | | | | | | | | | | | |
|---|---|---|---|---|---|---|---|---|---|---|---|
| | | fen | | | | | | | | (3.0) | (3.5) | (2008) |
| 8. | Lilla Backsjömyren | Mixed mire | Sweden | 62.41 | 14.32 | 1.6 | 563 | 8.5 | 125.2 (31.3) | 3.2 - 3.4 (3.3) | 1.5 - 2.2 (1.9) | Andersson and Schoning (2010) |
| 9. | Siikaneva | Boreal poor fen | Finland | 61.83 | 24.18 | 3.3 | 713 | 9.0 | 156.2 (17.3) | 2.6 - 2.7 (2.7) | 2.0 - 4.0 (3.0) | Aurela et al. (2007) |
| 10. | Ruosuo | Boreal poor fen | Finland | 65.65 | 27.32 | 1.0 | 650 | 9.3 | 135.4 (14.5) | 2.5 – 2.6 (2.5) | 1.9 - 2.8 (2.4) | Makila et al. (2001) |





**Table 5.** Observed dominant vegetation cover, long-term apparent rate of carbon accumulation
(LARCA), short-term net ecosystem exchange (NEE), and annual water table position (WTP)
compared with mean modelled values (1990-2000) for the 3 grid points in Scandinavian region

| Site (site no. in Table 4) | Fajemyr (5) | Degerö Stormyr (7) | Siikaneva (9) |
|---|---|---|---|
| **Dominant vegetation** | M, LSE, T | M, Gr | M, Gr, LSE |
| **Modelled Dominant vegetation** | M, LSE | M, Gr | M, Gr, LSE |
| **LARCA (g m$^{-2}$ yr$^{-1}$)** | 20-35 | - | 18.5 |
| **Modelled LARCA (g m$^{-2}$ yr$^{-1}$)** | 18.3 | 20.7 | 17.3 |
| **NEE (g m$^{-2}$ yr$^{-1}$) (period)** | 16.0-27.0 (2005-2006) | 12.9-16.7 (2001-2003) | 50.76-59.13 (2004-2005) |
| **Modelled NEE (g m$^{-2}$ yr$^{-1}$)** | 63.3 ± 17.1 | 30.6 ± 12.6 | 34.3 ± 28.0 |
| **WTP** | 0 to -20.0 | -4.0 to -20.0 | 2.0 to -25.0 |
| **Modelled WTP** | -15.2 ±1.83 | -2.9 ± 0.99 | 1.85 ± 0.42 |
| **Reference** | Lund et al. (2007) | Sagerfors et al. (2008) | Aurela et al. (2007) |





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
