# Peer review of "Modelling Holocene peatland dynamics with an individual-based dynamic vegetation model"

_Biogeosciences, 2016_

## Short Comment (SC1) · 28 Dec 2016

The hindcast is an important test to make sure that a model effectively captures past documented climate change and is realistic. Unfortunately, the simulated temperature and precipitation curves in Figure 5 show major discrepancies with reconstructed data from Canada and Scandinavia, challenging the robustness of the presented model.

1) Stordalen

The temperature history of northern Europe is well known and consists of cold and warm phases which alternate on a millennial scale. A good reconstruction for the last 2000 years is from Esper et al. 2014 (their Fig. 5c).

http://onlinelibrary.wiley.com/doi/10.1002/jqs.2726/abstract

The amplitude of the documented millennial-scale temperature changes is 2°C. Figure 5g in Chaudhary et al. does not include any of these millennial-scale changes, indicating that some important forcing parameter is missing or is significantly underestimated in the climate model that has been used.

Also, the longterm warming trend 4500-1700 yrs BP in Fig. 5g does not match with palaeoclimate reconstructions from Scandinavia. Following the mid Holocene climate optimum (8000-5000 years BP), a longterm-cooling trend has been observed. This is the exact opposite of what is shown in Fig. 5g in Chaudhary et al. The cooling trend is documented e.g. in Nesje et al. 2008 (their fig. 3; doi:10.1016/j.gloplacha.2006.08.004), Bjune 2005, Eldevik et al. 2014 http://www.sciencedirect.com/science/article/pii/S0277379114002650?np=y

Likewise, the precipitation history in Scandinavia was characterized by similar variability which is not reflected in Fig. 5i. For example, the Medieval Climate Anomaly (MCA) in Scandinavia was anomalously humid. Again, this is not reflected in Fig. 5i where the MCA appears rather dry.

2) Mer Bleu

Again, the temperature curve in Fig. 5h does not show any of the millennial-scale temperature variability. See e.g. Marchitto and deMenocal (2003) for a temperature reconstruction off Newfoundland for the past 4000 years. http://onlinelibrary.wiley.com/doi/10.1029/2003GC000598/abstract

Likewise, the longterm warming trend illustrated in Fig. 5h from 8000-3000 years BP does not match palaeoclimate reconstruction data which shows a cooling for Canada and the Arctic. See e.g. Gajewski 2015 http://www.sciencedirect.com/science/article/pii/S0921818115000417

The major discrepancies between simulated and reconstructed climate data cast doubt

on the skill of the model that has been used. How do the authors explain these discrepancies?

---

## Author Comment (AC1) · 25 Jan 2017

We thank the reviewer for these helpful comments.

The time series we adopted to force our model lack decadal and centennial scale variability, but capture inter-millennial trends (Figs. 1 and 2), as they were constructed by applying anomalies relative to present day from millennium time-slice experiments with the UK Hadley Centre Unified Model – HadSM3 (Pope et al., 2000). These reconstructions were evaluated and found broadly in line with proxy data for the mid-Holocene (6kyear cal. BP) in northern Europe and Canada (Muri, 2009). Detailed information is available here - http://www.climateprediction.net/projects/completed-project/mid-holocene/. The peat profile in both simulated localities, and across the northern

high latitudes in general, is a product of accumulation over millennia, and decadal to centennial-scale climatic variations may be assumed to have a minor influence on peat depth and density today.

The amplitude and variability in the millennial-scale temperature and precipitation smoothed out due to 10-year running mean, therefore we have included new figures here (see below Figs. 3 and 4). We found the applied temperature forcing is quite similar in magnitude and variability to Esper et al. (2014) (their Fig. 2 and 5c) for the last 2000 years (see Figs. 5 and 6 in this document) even though decadal and centennial variability is not captured.

Stordalen

The source of the discrepancy between proxy-based annual temperatures and the model-based data used as forcing in our study is due to errors in the modelled winter (October-March) temperatures (Fig. 7 b and d). While this could have some impact on our simulations, summer temperatures (Figs. 7 a and c) are more important than winter temperature for the peat growth in our model. This is evident from Fig. 8 (in this document) where winter anomalies from October-March were not added to the observed surface temperature and a very limited effect was noticed on the peat accumulation trajectory compared to STD experiment. In contrast, when we set the summer temperature anomalies to zero, the peat accumulation differs compared to STD experiment. Although the upward trend in mean annual temperature is influenced by winter anomalies (Oct-Mar), that discrepancy didn't affect the overall peat growth. Hence we would argue that the climate data used as forcing in our study are sufficiently realistic in terms of their influence on modern peat amount.

Mer Bleue

Mean annual temperature has been relatively stable for the last 8500 years in Ontario region. There was no warming and cooling trend in the proxy-based climate reconstruction (Muller et al. 2003, Page 65 of their paper) and the same dataset was used

to model peat accumulation at Mer Bleue by Frolking et al. (2010) (see Page 8 of their paper). Kaufman et al. (2004) showed (in their Fig. 7) that the Holocene Thermal Maximum (HTM) initiated around 6-7 kyears cal. BP and terminated around 3-4 kyears cal. BP near Hudson Bay and this is reflected in the Holocene time-series of temperature used as forcing in our study (Fig. 2b in this document). The HTM was delayed in this region due to remnants of Laurentide ice sheet and the period was warmer than today by 1-2 degree C which coincides with vegetation density and northern advance of arctic tree line (Page 238, Rolland et al. 2008).

While we maintain that the climate forcing data used in our study are defensible and adequate for representing the main climatic drivers of peatland development at our study sites, there are obviously uncertainties in reconstructions of past climate, whether using proxy data or models. We have added some more explanation on the potential effects of climate uncertainty on the model simulations around Line 467-476 of our paper.

———————————————————

[Figure]

**Fig. 1.** Applied annual temperature (additive) and precipitation (multiplicative) anomalies for the Stordalen.

**Fig. 2.** Applied annual temperature (additive) and precipitation (multiplicative) anomalies for the Mer Bleue.

[Figure]

[Figure]

**Fig. 3.** Applied temperature (°C) and precipitation (mm/day) forcing for the last 4700 years at Stordalen

**Fig. 4.** Applied temperature (°C) and precipitation (mm/day) forcing for the last 8400 years at Mer Bleue

[Figure]

[Figure]

**Fig. 5.** Comparison between Esper et al. 2014 (Fig. 2a - N-Scandinavia/Finnish Lapland) and applied temperature (°C) forcing for the last 2000 years at Stordalen.

[Figure]

[Figure]

[Figure]

**Fig. 6.** Comparison between Esper et al. 2014 (Fig. 5c - northern Europe) and modelled June-July-August (JJA) temperature (°C) and precipitation (mm/day) at Stordalen

**Fig. 7.** Applied annual temperature anomalies for (a) summer months (June-August) and (b) winter months (October-March) and mean annual (c) summer and (d) winter temperature anomalies for the Stordalen.

**Fig. 8.** Peat depth sensitivity to applied temperatures anomalies. No anomalies added to the mean monthly-observed temperature (1913-1942) (ALL = 0), no winter temperature anomalies applied (Mar-Oct = 0) and n

---

## Author Comment (AC2) · 26 Jan 2017

References:

Esper, J., Duthorn, E., Krusic, P. J., Timonen, M., and Buntgen, U.: Northern European summer temperature variations over the Common Era from integrated tree-ring density records, Journal of Quaternary Science, 29, 487-494,doi: 10.1002/jqs.2726, 2014.

Frolking, S., Roulet, N. T., Tuittila, E., Bubier, J. L., Quillet, A., Talbot, J., and Richard, P. J. H.: A new model of Holocene peatland net primary production, decomposition, water balance, and peat accumulation, 1 Article, Earth System Dynamics, 1-21 pp., 2010.

Kaufman, D. S., Ager, T. A., Anderson, N. J., Anderson, P. M., Andrews, J. T., Bartlein,

P. J., Brubaker, L. B., Coats, L. L., Cwynar, L. C., Duvall, M. L., Dyke, A. S., Edwards, M. E., Eisner, W. R., Gajewski, K., Geirsdottir, A., Hu, F. S., Jennings, A. E., Kaplan, M. R., Kerwin, M. N., Lozhkin, A. V., MacDonald, G. M., Miller, G. H., Mock, C. J., Oswald, W. W., Otto-Bliesner, B. L., Porinchu, D. F., Ruhland, K., Smol, J. P., Steig, E. J., and Wolfe, B. B.: Holocene thermal maximum in the western Arctic (0-180 degrees W), Quaternary Science Reviews, 23, 529-560,doi: 10.1016/j.quascirev.2003.09.007, 2004.

Muller, S. D., Richard, P. J. H., Guiot, J., de Beaulieu, J. L., and Fortin, D.: Post-glacial climate in the St. Lawrence lowlands, southern Quebec: pollen and lake-level evidence, Palaeogeography Palaeoclimatology Palaeoecology, 193, 51-72,doi: 10.1016/S0031-0182(02)00710-1, 2003.

Muri, H.: Evaluating forcings in an ensemble of paleo-climate models., D.Phil thesis, University of Oxford, Dept. of Physics. , 2009.

Pope, V. D., Gallani, M. L., Rowntree, P. R., and Stratton, R. A.: The impact of new physical parametrizations in the Hadley Centre climate model: HadAM3, Clim. Dyn., 16, 123-146,doi: 10.1007/s003820050009, 2000.

Rolland, N., Larocque, I., Francus, P., Pienitz, R., and Laperriere, L.: Holocene climate inferred from biological (Diptera : Chironomidae) analyses in a Southampton island (Nunavut, Canada) lake, Holocene, 18, 229-241,doi: 10.1177/0959683607086761, 2008.

---

## Short Comment (SC2) · 27 Jan 2017

I would like to thank the authors for their detailed reply. I understand that millennial-scal variability was not the objective of the simulations, even though it would have been useful that such important climate changes were included in the model, in my opinion.

Unfortunately, climate models have also struggled to reproduce Holocene long-term trends, as documented by Marcott et al. 2013 (DOI: 10.1126/science.1228026). Longterm trends of reconstructed vs. simulated temperatures show major discrepancies. I am attaching Figure S8 from the Supplement of the Marcott et al. paper which shows the simulated temperatures. Please compare to Figure S12 (also attached) which presents the reconstructed temperatures. These look very different to me.

[Figure]

**Fig. S8**: Simulated global mean temperature for the last 11000 years at the 73 proxy sites (black) from the ECBilt-CLIO transient simulations (*81*), and the global mean temperature assuming a seasonal proxy bias (red) as described in text.

**Fig. 1.**

[Figure]

Fig. S12: Temperature reconstructions using multiple time-steps. **(a)** Global temperature envelope (1-σ) (light blue fill) and mean of the standard temperature anomaly using a 20 year interpolated time-step (blue line), 100 year time-step (pink line), and 200 year time-step (green line).  Mann et al.'s (*2*) global temperature CRU-EIV composite (darkest gray) is also plotted.  Uncertainty bars in upper left corner reflect the average Monte Carlo based 1σ uncertainty for each reconstruction, and were not overlain on line for clarity.  **b** same as **a** for the last 11,300 years. Temperature anomaly is from the 1961-1990 yr B.P. average after mean shifting to Mann et al.(*2*).

**Fig. 2.**

---

## Referee Comment (RC1) · Anonymous Referee #3 · 21 Feb 2017

Chaudhary and coworkers adapted the LPJ-GUESS model to simulate peatland and permafrost dynamics at boreal peatland sites. The model can dynamically grown peat producing a peat depth that can be compared to observations. The model additionally has the ability to simulate water transfer between sub-grid patches that are of differing heights. After parameterizing the model at one site they ran it at nine more as well as running the initial site through two RCP scenarios.

The paper is generally well presented with reasonable figures. While the model does appear able to produce some reasonable peat depths at some sites, it fails at others. This along with the rather poor NEE performance makes me wonder about how well the

model is truly able to provide predictions of changes in the C cycle in future simulations. I think more effort should be place on demonstrating the model performance before attempting future scenarios. I think many resources for model validation were not used that could have been. I recommend revisions before this MS can be published.

General comments:

Looking at figure 8, it is apparent that the modelled peat depth vs. the observed peat depth is not great. This I can understand, the climate of the holocene when these peatlands were forming is not likely to be well captured with climate and conditions as they were able to produce. My main issue is the NEE estimates from the model are also not corresponding well to observations. In this instance the conditions at the sites are well known and reasonable climate should be possible. The problem with the NEE values being off significantly is that it is difficult then to trust when the model predicts the peat depth at sites should grow significantly or shrink since the NEE is how that is controlled in essence.

I also feel that many of the model outputs are not compared to observations when they should be. For e.g., the active layer depth is only compared at one of the 10 sites simulated. Do any of the other sites have information about ALD? Do any have ALD timeseries for comparision? What about the PFT distribution. The PFT distribution is shown in Table 5 but is just a presence or absence. Is there any more quantitative values that can be used to compare the model to obs here? Given the productivity differences between PFTs, it could be instructive for interpretation of model-obs differences. For the WTP, could there be some comparisons not just of some mean annual value but of the timeseries? Is the water table correct at the different times of the year? In general, much of the model performance is sort of dumped into tables, since this is the first paper describing this peatland version of LPJ-GUESS I believe more effort has to be put into demonstrating that the model doesn't get things 'sort of ok' for the wrong reasons.

Specific comments:

line 10: Change 'current' to 'many' in the start of the second sentence. Other models do indeed have peatlands, e.g.

Wu, Y., Verseghy, D. L. and Melton, J. R.: Integrating peatlands into the coupled Canadian Land Surface Scheme (CLASS) v3.6 and the Canadian Terrestrial Ecosystem Model (CTEM) v2.0, Geoscientific Model Development, 9(8), 2639–2663, 2016.

Alexandrov, G. A., Brovkin, V. A. and Kleinen, T.: The influence of climate on peatland extent in Western Siberia since the Last Glacial Maximum, Sci. Rep., 6, 24784, 2016.

Kleinen, T., Brovkin, V. and Schuldt, R. J.: A dynamic model of wetland extent and peat accumulation: results for the Holocene, Biogeosciences, 9(1), 235–248, 2012.

Stocker, B. D., Spahni, R. and Joos, F.: DYPTOP: a cost-efficient TOPMODEL implementation to simulate sub-grid spatio-temporal dynamics of global wetlands and peatlands, Geoscientific Model Development, 7(6), 3089–3110, 2014.

l. 30: Do you really mean Wania et al here? That was a modelling study... If you are talking about the mask used for the peatland regions that was Tarnocai, not Wania. Cite the true reference please.

l. 38: Could add some of the refs I gave above to this list.

l. 40: See the Stocker ref along with Alexandrov to see if this statement is correct still.

l. 43: 'northern high latitutdes, ... , could' - suggest adding some commas.

l. 70: By soil resources, you mean water right? nutrients are not simulated in this version, correct?

l. 80: So how many soil layers? This description in this paragraph is different than the figure. Please make them more congruent. I am still not sure how many layer were truly simulated.

l. 95 : based on what studies?

l.97: I don't understand the 'fresh litter debris decomposes through surface forcing until the last day of the year'. Surface forcing?

l 117 : Please put the values of all these constants in the text and not just the table. It was confusing until I found the table since the table is not really mentioned until much later.

l 117: How does K relate to $K_o$ or $K_i$?

Eqns 4 and 5 - would be nice if these were plotted, easier than trying to imagine in the head...

All eqns - be consistent between 1.0 and 1 etc. in the equations.

Eqn 6 - units?

l. 153 - value of the min and max bulk densities? Calculated somewhere?

pg 7 - choose one: cm or mm and please stick to whichever is chosen.

Eqn 8 - Did I miss how F was found?

Eqn 12 : Are you sure this is a change of porosity? This looks more like a fraction of original porosity. Change to me implies something like flux.

L 227: So moss can get water from 50 cm mineral + peat depth until peat => 50 cm? This seems strange and would greatly advantage moss for quite a while. Is there any indication that moss can access water almost 1 m down? I find this difficult to believe.

l. 253: How are the heights done? Is this peat height or actual elevation?

l. 316: Sure it conserves the IAV - but it also then pegs the IAV as the same for the whole simulation instead of perhaps changing through time.

l. 318: No, it is really reanalysis or interpolated climate. There are no 'observed'

gridded products available.

l. 341: Can you please expand more on why you needed to keep the mineral and peat layers saturated during initialization. This to me would imply that your model was out of equilibrium at the start of your runs and thus the transient behaviour would be influenced by the model initial conditions. This is a bit worrying. Once you released the saturated conditions the model could then over-react to dry conditions as mentioned.

l. 349 : This comment about adjusting to the local WTP really drives my request for comparing timeseries of WTP since it is then apparent that we cannot put too much stock in the mean WTP values matching reasonably.

l. 400: 'lower than 50 kgm-3' - higher meant?

l. 416: Any obs to compare with here?

l. 421: Are there any vegetation reconstructions available for these sites? Pollen cores that can help determine if the model successional sequence is reasonable?

Fig 1: why are the mosses all different colours? Can this diagram be simplified - like only a couple grass instead of that dark mat? Should permafrost maybe be 'frozen soil' or maybe distinguish seasonally frozen soil from perenially frozen? Why is the permafrost bubble circular? Would the model really have a different bottom permafrost depth between its tiles in the same gridcell? I can understand a different top depth but not really a bottom.

Fig 2: Perhaps choose a different acronym than UM since that is also used in the MS to talk about a model.

Fig 6: I find the acronym choice non-sensible. Why does the final S of deciduous shrubs be S and not a D? Not a big deal but it makes it harder to quickly remember what the acronym stands for.

Fig 7: No description of the X and Z in the caption. What do Top, Middle, and Bottom

really correspond to? This gets back to my earlier comment that I don't understand how your soil layers were divided.

Fig 8: As I said in the general comments, this figure does not give much confidence when combined with the NEE results.

Fig A1 - perhaps add total water (liquid and frozen) so we can see if the total content was changing and it wasn't just changing phase.

Table 2: density is needing the o as an subscript. Also please bring these all into the main text, it is annoying to have to search out the table when one is reading the text (and it is often not mentioned that one needs to search for a table...)

Table 5: WTP units? Please put in proportions of the veg so we can tell if the proportions modelled are in any way correct rather than just presence/absence.
* * *

---

## Referee Comment (RC2) · A.J. Baird (Referee) · 21 Feb 2017

The comment was uploaded in the form of a supplement:
http://www.biogeosciences-discuss.net/bg-2016-319/bg-2016-319-RC2-supplement.zip

---

## Author Comment (AC3) · 21 Mar 2017

We appreciate the time and effort spent by the reviewers in reviewing this manuscript. We have addressed all the issues indicated in the review reports and believe that the revised version will meet the journal's publication requirements.

General comments: Looking at figure 8, it is apparent that the modelled peat depth vs. the observed peat depth is not great. This I can understand, the climate of the holocene when these peatlands were forming is not likely to be well captured with climate and conditions as they were able to produce. My main issue is the NEE estimates from the model are also not corresponding well to observations. In this instance the conditions at the sites are well known and reasonable climate should be possible. The problem

with the NEE values being off significantly is that it is difficult then to trust when the model predicts the peat depth at sites should grow significantly or shrink since the NEE is how that is controlled in essence.

I also feel that many of the model outputs are not compared to observations when they should be. For e.g., the active layer depth is only compared at one of the 10 sites simulated. Do any of the other sites have information about ALD? Do any have ALD timeseries for comparision? What about the PFT distribution. The PFT distribution is shown in Table 5 but is just a presence or absence. Is there any more quantitative values that can be used to compare the model to obs here? Given the productivity differences between PFTs, it could be instructive for interpretation of model-obs differ-ences. For the WTP, could there be some comparisons not just of some mean annual value but of the timeseries? Is the water table correct at the different times of the year? In general, much of the model performance is sort of dumped into tables, since this is the first paper describing this peatland version of LPJ-GUESS I believe more effort has to be put into demonstrating that the model doesn't get things 'sort of ok' for the wrong reasons.

Response: We agree with the reviewer's point that we need to give a better demonstra-tion of the skill of our model. We have therefore clarified and improved these aspects of the paper, and here we provide a summary of our changes.

Net Ecosystem Exchange (NEE)

We ran the model with the observed dataset for the Stordalen site from the year 2001-2012 and our model predicts reasonable NEE values for the Stordalen site (see Fig. 11 in the revised manuscript (RM)). NEE outputs for the other three sites are almost within the range of observed NEE values (see Table 5 in the RM), albeit with some differences. Fajemyr and Degerö Stormyr are disturbed (i.e. subject to anthropogenic influence), which we have haven't accounted for in the model but relatively less influenced sites (Stordalen and Siikaneva) showed close match with the observed values. Furthermore,

water borne carbon fluxes are not included in the model and that is also one of the potential causes of this discrepancy.

Peatlands are heterogeneous ecosystems and the carbon fluxes vary spatially and temporally within the landscape. Ecosystem scale NEE can be obtained using eddy flux towers, but previous studies have highlighted that peatland short-term NEE fluxes show a lot of variability and may not be indicative of long-term peatland behavior (Lafleur et al., 2003; Aslan-Sungur et al., 2016). We believe that it is equally important for models to capture the long-term carbon accumulation rate (LARCA). We find our LARCA values are quite close to the Fajemyr and Siikaneva sites and in our companion paper (Biogeosciences Discuss., doi:10.5194/bg-2017-34, 2017) we have demonstrated that the model was able to capture the right LARCA values in almost all the major peatland regions across the Arctic.

Active Layer Depth (ALD)

We have compared the simulated annual ALD with the observed values (1990-2011) for the Stordalen site (see Fig. 9 in the RM) and even analysed the hummocks and hollow ALD separately. We found that the magnitude, variability and trend of the simulated annual ALD are close to the observed values (see Fig. 9 in the RM). ALD is shallow in drier, elevated areas while deeper in wetter hollows, a phenomenon observed in many permafrost peatland sites (Johansson et al., 2013). The ALD trends over the observed period are also similar. Observed ALD trends are 69.2 cm/year, whereas the modelled ALD trend is 68.2 cm/year over the same period.

Furthermore, in our companion paper we produced a permafrost extent map (see Fig. 5 on the page 28 in the companion paper) that captures the main features of the permafrost distribution map developed by Tarnocai et al. 2009 (see their Fig. 1 on page 3), highlighting the robustness of the model in predicting the existence of permafrost in other areas besides the sites discussed in this paper.

Water Table Position (WTP) and PFT distribution

We have compared the observed annual and monthly WTP for a semi-wet patch in Stordalen to the simulated result with our model's semi-wet patches for the period 2003-12. The results were quite consistent with observed values (see Figs. 8 and A5 in the RM).

For majority of Stordalen peatland history, different species of mosses occupied the mire. The model predicted correctly the dominance of wet PFT during 4000-3000 cal. BP (see Fig. 6a in the RM). However, there was a certain period between 700-1700 cal. BP when graminoids were again the dominant PFT, but we could not reproduce that period due to the climate forcing used here, as explained in the text. See lines: 527-533 in the RM.

We have given further responses to each of the reviewer's comments on WTP, plant distribution, bulk density and ALD below.

line 10: Change 'current' to 'many' in the start of the second sentence. Other models do indeed have peatlands, e.g.

Response: We agree with the reviewer on this but many current models do not have a multiple peat layer representation with permafrost functionality and we wanted to highlight such functionality in current dynamic global vegetation models (DGVM), not in other models. Only Kleinen et al. (2012) and Stocker et al. (2014) have introduced initial representations of peat formation in a DGVM framework but both of them lack permafrost functionality. The papers by Wu et al. (2016) and Alexandrov et al. (2016), however, describe quite recent model developments but both were published last year when this study was being submitted, so we couldn't refer them. Also, these two models are not DGVMs (see Table S1 on page 45 (line: 793) in the RM- there are many other models apart from the one mentioned in this table). In Table S1, comparison of the functionality and scope of a representative set of current peatland models have been mentioned. So, although we think current is a more appropriate word than many, we have modified the earlier sentence and clarified the above points for the readers.

Revised text (lines: 11-13 in the RM) - However, most DGVMs do not yet have detailed representations of permafrost and non-permafrost peatlands, which are an important store of carbon particularly at high latitudes.

l. 30: Do you really mean Wania et al here? That was a modelling study... If you are talking about the mask used for the peatland regions that was Tarnocai, not Wania. Cite the true reference please.

Response: We have now modified the sentence and added that reference (see lines: 30-31 in the RM).

Revised text: Around 19% (3556 × 103 km2) of the soil area of the northern peatlands coincides with low altitude permafrost (Tarnocai et al., 2009; Wania et al., 2009a).

l. 38: Could add some of the refs I gave above to this list.

Response: We have added a reference to Stocker et al. (2014) (see lines: 37-39 in the RM). Here we are referring specifically to DGVMs.

Revised text: Only a few DGVMs include representations of the unique vegetation, biophysical and biogeochemical characteristics of peatland ecosystems (Wania et al., 2009a, b; Kleinen et al., 2012; Tang et al., 2015).

l. 40: See the Stocker ref along with Alexandrov to see if this statement is correct still.

Response: Many models described in these references did not have multiple annual layer representations of peat accumulation and decomposition so we haven't included them in this sentence (see Table S1 in the RM (line: 793)). However, we have reformulated the sentence for more clarity (see lines: 35-45 in the RM) and added a separate sentence to acknowledge the work done by other modelling groups. Revised text- Dynamic global vegetation models (DGVMs) are used to study past, present and future vegetation patterns from regional to global scales, together with associated biogeochemical cycles and climate feedbacks, in particular through the carbon cycle (Smith et al., 2001; Friedlingstein et al., 2006; Sitch et al., 2008; Strandberg et al., 2014;

Zhang et al., 2014). Only a few DGVMs include representations of the unique vege-
tation, biophysical and biogeochemical characteristics of peatland ecosystems (Wania
et al., 2009a, b; Kleinen et al., 2012; Tang et al., 2015). Model formulations of mul-
tiple peat layer accumulation and decay have been proposed and demonstrated at
the site scale (Frolking et al., 2010; Heinemeyer et al., 2010) but have not yet, to our
knowledge, been implemented within the framework of a DGVM. However, peatland
processes are included in some other types of model frameworks (Morris et al., 2012;
Alexandrov et al., 2016; Wu et al., 2016) and been shown to perform reasonably for
peatland sites. Large area simulations of regional peatland dynamics have been per-
formed by (Kleinen et al., 2012; Schuldt et al., 2013; Stocker et al., 2014; Alexandrov
et al., 2016) (see Table S1).

l. 43: 'northern high latitudes, ... , could' - suggest adding some commas.

Response: Thank you, we have done this (see lines: 47-49 in the RM)

Revised text: Current climate models predict that the northern high latitudes, where
most of the peatlands and permafrost areas are present, could experience warming of
more than 5°C by 2100 (Hinzman et al., 2005; Christensen et al., 2007; IPCC, 2013).

l. 70: By soil resources, you mean water right? nutrients are not simulated in this
version, correct? Response: Yes, by soil resources we mean water, not nutrients, and
we have clarified this in the text (see lines: 76-79 in the RM)

Revised text: Vegetation structure and dynamics follow an individual- and patch-based
representation in which plant population demography and community structure evolve
as an emergent outcome of competition for light, space and soil water among simulated
plant individuals, each belonging to one of a defined set of plant functional types (PFTs)
with different functional and morphological characteristics (see below).

l. 80: So how many soil layers? This description in this paragraph is different than the
figure. Please make them more congruent. I am still not sure how many layer were

truly simulated.

Response: For Stordalen, 4739 + 100 peat layers were simulated, i.e. one peat layer for each of the 4739 years after inception until year 2000, followed by a 100-year projection from 2001 to 2100. For Mer Bleue, it was 8400 + 100 layers. We can't show that many layers in a figure so we simplified the representation in schematic representation of the model (see Fig. 1 in the RM and lines: 91-98).

Revised text: A one-dimensional soil column is represented for each patch (defined below), divided vertically into four distinct layers: a snow layer of variable thickness, one dynamic litter/peat layer of variable thickness corresponding to each simulation year (e.g. 4739 + 100 layers by the end of the simulations, described in Section 2.4 below, for Stordalen), a mineral soil column with a fixed depth of 2 m consisting of two sublayers: an upper mineral soil sublayer (0.5 m) and a lower mineral soil sublayer (1.5 m), and finally a "padding" column of 48 m depth (with 10 sublayers) allowing the simulation of accurate soil thermal dynamics (Wania et al., 2009a). The insulation effects of snow, phase changes in soil water, precipitation and snowmelt input and air temperature forcing are important determinants of daily soil temperature dynamics at different depths.

l. 95 : based on what studies?

Response: We added the references after the second sentence but now we have moved it a little further in the text for clarity (see lines: 104-105 in the RM)

Revised text: Woody litter mass from shrubs decomposes relatively slowly because it is made up of hard cellulose and lignin (Aerts et al., 1999; Moore et al., 2007).

l.97: I don't understand the 'fresh litter debris decomposes through surface forcing until the last day of the year'. Surface forcing?

Response: When the litter (leaves and stems, where appropriate) is dropped on the ground surface it doesn't become a part of peat column (formed of multiple layers –

see above) instantaneously. This litter then decomposes at rates depending on the surface conditions in that year, i.e. surface temperature and moisture, becoming the top layer in the peat layer of the soil column the following year. So, in our framework, we decompose the litter mass present on the peat surface for the first year before it transforms into a peat layer. However, for dead roots, we add them directly to the peat layers where they belong (see lines: 106-110 in the RM).

Revised text: Fresh litter debris decomposes on the surface through exposure to surface temperature and moisture conditions until the last day of the year. The decomposed litter carbon is assumed to be released as respiration directly to the atmosphere while any remaining litter mass is treated as a new individual peat layer from the first day of the following year, which then underlies the newly accumulating litter mass.

l 117 : Please put the values of all these constants in the text and not just the table. It was confusing until I found the table since the table is not really mentioned until much later.

Response: We have included constant values at all the appropriate places and also referred earlier to Table 2 in the revised manuscript. See lines: 150-153, 156-158, 166-168, 219, 222, 231 and 234 and Table 2 in the RM.

l 117: How does K relate to K_o or K_i?

Response: We use K in general terms to refer to an overall decomposition rate of the entire peat column. $k_i$, however, is the decomposition rate of an individual peat layer (i) (see Eq. 2 and 3 in the RM) and $k_o$ is the initial decomposition rate. This distinction was introduced by Clymo (1984) and is also used in the Frolking et al. (2010) and some other publications on peatland modelling. We explain these variables just below the equations where they were first defined and there we also referred to these papers (see lines: 129-140 and 144 in the RM)

Eqns 4 and 5 - would be nice if these were plotted, easier than trying to imagine in the

head...

Response: We agree with the reviewer. We have now included them as a new figure (Fig. A1 in the RM; see lines: 152, 158 and 728 in the RM)

In Fig. A1 (in the RM), we presented assumed decomposition dependency on (a) soil temperature and (b) soil water content.

All eqns - be consistent between 1.0 and 1 etc. in the equations.

Response: Agreed. We have changed all equations to use 1 consistently

Eqn 6 - units?

Response: kg m-3 (Included in the text see lines: 166-167 in the RM)

l. 153 - value of the min and max bulk densities? Calculated somewhere?

Response: Minimum bulk density – 40 kg m-3 and maximum bulk density - 120 kg m-3 They are prescribed values and inspired by the work of Frolking et al. (2010) (see their Fig. 3 and Table 2 on page 6) where they used a similar range of 30-120 kg m-3. Heinemeyer et al. (2010) prescribed a range of 50-100 kg m-3 (see their section 2.3.5 on page 214). Similar ranges can be found in the majority of peatlands (see the explanation below).

pg 7 - choose one: cm or mm and please stick to whichever is chosen.

Response: Agreed. We have changed it to cm throughout.

Eqn 8 - Did I miss how F was found?

Response: We gave the reference in the beginning but now we have placed it right next to F (see line: 216 in the RM). F is the fraction of the modelled area subject to evaporation (i.e. bare soil fraction) calculated by LPJ-GUESS, and explained in Gerten et al. (2004) – see Eq. 9 on Page 254 in their paper. Revised text: Evaporation can only occur when the snowpack is thinner than 1 cm and is calculated following the

approach of Gerten et al. (2004), as in the standard version of LPJ-GUESS:

ET= 1.32 .E .W_c^2. F

where E is the climate-dependent equilibrium evapotranspiration (cm), Wc is the water content on the top 10 cm of the peat soil and F is the fraction of modelled area subject to evaporation, i.e. not covered by vegetation (Gerten et al., 2004).

Eqn 12: Are you sure this is a change of porosity? This looks more like a fraction of original porosity. Change to me implies something like flux.

Response: Porosity, the volume of empty spaces over the total volume, varies between 0-1. In our implementation, porosity is a function of bulk density and it decreases from (1-40/800 = 0.95) to (1-120/800 = 0.85) as bulk density increases. Frolking et al. (2010) used a similar function (see their Eq. 18 on page 7) and we have given a reference to it. Ryden et al. (1980) found a similar observed range of 0.97-0.88 in depressed patches and 0.93-0.87 in elevated areas in Stordalen (see their Tables 2 and 3 on page 37-39).

L 227: So moss can get water from 50 cm mineral + peat depth until peat => 50 cm? This seems strange and would greatly advantage moss for quite a while. Is there any indication that moss can access water almost 1 m down? I find this difficult to believe.

Response: In our representation, moss can only take up water from the top 50 cm of the soil. It can take up water from the top 50 cm of the mineral soil during the spin up phase, after which it starts taking up the water from the peat soil (but again only the top 50 cm). We have clarified this in the text. See lines: 243-249 in the RM.

Revised text: In the beginning of the peat accumulation process, plant roots are present both in peat and upper and lower mineral soil layers but their mineral soil root distribution declines linearly as peat grows (see Fig. 2) and the corresponding mineral layer reduction is used to access water from the peat layers. Mosses are assumed only to take up water from the top 50 cm of the mineral soil in the beginning but once the peat depth exceeds 50 cm they only take water from the peat layers (top 50 cm of the peat

layer). Other PFTs can continue to take up water both from the mineral and peat soils until peat depth reaches 2 m, and from only from the peat soil thereafter.

l. 253: How are the heights done? Is this peat height or actual elevation?

Response: It is the cumulative peat height minus the initial elevation.

l. 316: Sure it conserves the IAV - but it also then pegs the IAV as the same for the whole simulation instead of perhaps changing through time.

Response: We agree with the reviewer on this point but it is a common technique used for reconstruction of palaeo climate forcing when there are no proxy based climate data available from which one could infer a change in IAV. We have clarified the text to explain the effects of not capturing the IAV, see lines: 507-515 and 527-533.

Revised text: Studies of the influence of GCM-generated climate uncertainty (i.e. variations in climate output fields among GCMs) on carbon cycle model prediction, underline the high prediction error that can arise, for example in present-day biospheric carbon pools and fluxes (Ahlström et al., 2013; Anav et al., 2013; Ahlström, 2016). Potential bias and errors in the predicted climate may be expected to be even higher in palaeoclimate simulations, not least due to the absence of instrumental observations for validating the models. Furthermore, in this study additional bias could arise due to the interpolation procedure used to transform GCM output fields into monthly anomalies, required to force our model. These were generated by linearly interpolating between the climate model output, which is only available at 1000-year intervals. As such, the applied anomalies do not capture decadal or centennial climate variability that can contribute to climate-forced variable peat accumulation rates and vegetation dynamics on these timescales (Miller et al., 2008).

Mosses emerged as the dominant PFT at the beginning of the simulation, while 300-400 years after peat inception shrubs started establishing in the higher elevated patches as a result of a lowering of WTP. Graminoids were not productive during the
entire simulation period apart from the period 4-3kyr cal. BP (Kokfelt et al., 2010). The model predicted correctly the dominance of graminoids, characteristic of wet conditions, during 4-3kyr cal. BP. However, a period of graminoid dominance between 700-1700 cal. BP was not accurately captured. One explanation can be the absence of decadal and centennial climate variability in the adopted climate forcing data, resulting in an "averaging out" of moisture status over time that elminates wet episodes needed for graminoids to be sufficiently competitive.

l. 318: No, it is really reanalysis or interpolated climate. There are no 'observed' gridded products available.

Response: Yang et al. (2012) developed an observed climate time series (50 m resolution) from 1913-2006 for the Stordalen catchment. We used the first 30 years (1913-1942) mean ($\mu$) and standard deviation ($\sigma$) and drew randomly generated climate data assuming a normal distribution. The randomly generated climate data is then applied to the relative anomalies derived from the gridcell nearest to the location of the site from millennium time-slice experiments using the UK Hadley Centre's Unified Model. Explained in detail between lines – 350-362.

Existing Text: The high spatial resolution (50 m), modern observed climate dataset was developed by Yang et al. (2012) for the Stordalen site. In this dataset, the observations from the nearest weather stations and local observations were included to take into account the effects of the Torneträsk lake close to the Stordalen catchment. The monthly precipitation data (1913-2000) for Stordalen at 50 m resolution were downscaled from 10 min resolution using CRU TS 1.2 data (Mitchell and Jones, 2005), a technique quite common for cold regions (Hanna et al., 2005). The precipitation data was also corrected by including the influences of topography and also by using historical measurements of precipitation from the Abisko research station record. Finally, monthly values of Holocene temperature were interpolated to daily values, monthly precipitation totals were distributed randomly among the number (minimum 10) of rainy days per month from the climate dataset and the monthly CRU values of cloudiness for the

first 30 years from the year 1901-1930 were repeated for the entire simulation period. We added random variability to the daily climate values by drawing random values from a normal distribution with monthly mean ($\mu$) and standard deviation ($\sigma$) of the monthly observed climate were used for Stordalen from the period of 1913-1942 and for Mer Bleue, 30 years of monthly CRU values from the period of 1901-1930 were used.

l. 341: Can you please expand more on why you needed to keep the mineral and peat layers saturated during initialization. This to me would imply that your model was out of equilibrium at the start of your runs and thus the transient behaviour would be influenced by the model initial conditions. This is a bit worrying. Once you released the saturated conditions the model could then over-react to dry conditions as mentioned.

Response: We adopted this model initialisation strategy partially to avoid any sudden collapses of the peat column in very dry conditions because young, shallow peat can become drier or wetter within a very short time span and continuous dry periods would increase temperature dependent decomposition rates and reduce the accumulation rate markedly. Furthermore, peatlands develop due to the complex processes of terrestrialisation or plaudification that are not fully captured by our model in its current form. We agree that keeping the patches wet enough during the initialization phase is a limitation of our model, but it is one that corresponds to allowing peat growth in locally, low-lying saturated ecosystems in each gridcell.

l. 349 : This comment about adjusting to the local WTP really drives my request for comparing timeseries of WTP since it is then apparent that we cannot put too much stock in the mean WTP values matching reasonably.

Response: We forced the model with observed climate from 2001-2012 and found modelled annual and monthly WTP for semi-wet patches are quite close to the observed annual WTP from the year 2003 to 2012 (see Figs. 8 and A5 in the RM and lines: 538-548 in the RM).

Revised text: The modelled annual and monthly WTP from 2003-2012 in semi-wet

patches and modelled annual ALD 1990-2012 is in good agreement with the observed values for the Stordalen region (Figs. 8, 9 and A5) supporting the ability of model to capture hydrological dynamics that further drive peatland dynamics. For the additional evaluation sites, modelled dominant vegetation cover, LARCA and WTP were in good agreement with the observed values for the three selected sites at which this information was available. Under the present climate, Stordalen was simulated to be a small sink for atmospheric $CO_2$, in agreement with observed NEE (see Fig 11). NEE inter-annual range is likewise close to observations for the other Scandinavian sites (Table 5). However it is uncertain whether recent annual observations of NEE necessarily reflect the long-term peatland carbon balance, in view of high variability on multiple timescales. For example, Fajemyr has switched between source (14.3-21.4 g C m-2 yr-1 in 2005-2006; 23.6 g C m-2 yr-1 in 2008) and sink (-29.4 g C m-2 yr-1 in 2007; -28.9 g C m-2 yr-1 in 2009) conditions in recent years, and this variability has been attributed to disturbances and intermittent drought conditions (Lund et al., 2012).

In Fig. 8 (RM) (a) the total sum of precipitation (in cm) and (b) a comparison between observed and simulated mean annual WTP for semi-wet patches in Stordalen for the period 2003-2012 was presented In Fig. A5 in the RM a comparison between observed and simulated monthly mean WTP (JJA) for semi-wet patches in Stordalen for the period 2003-2012 have been shown.

l. 400: 'lower than 50 kg m-3' - higher meant? l. 416: Any obs to compare with here?

Response: Simulating bulk density is a challenge. In some peatlands, it may increases with depth due to compaction (Clymo, 1991; Novak et al., 2008) but other studies have shown no net increase in the bulk density with depth in some other locations (Tomlinson, 2005; Baird et al., 2016). In our study, the simulated bulk density is a function of the total mass remaining and in the peat profile it varies between 40-102 kg m-3 for Stordalen. Ryden et al. (1980) given a range of 45-230 kg m-3 (see page 41 and Table 5 and 6 in their paper) and our values are well within this range. We also find bulk density doesn't decline with depth and it is highly variable down the profile.

Since the lower layers were frozen, they didn't decompose significantly and their bulk densities remain higher relative to other partially frozen or unfrozen layers. The value referred to by the reviewer is the mean value of the entire simulated peat profile and it was lower than 50 kg m-3 since the majority of peat layers are not highly compacted as a result limited decomposition due to permafrost or high water contents (see lines: 429:440 in the RM).

Revised text: When the peat layers had decomposed sufficiently and lost more than 70% of their original mass (Mo), their bulk density increased markedly. The observed monthly and annual WTP for the semi-wet patches and mean annual ALD were very near to the simulated values (see Figs. 8, 9 and A5). The simulated bulk density varies between 40-102 kg m-3 and the mean annual bulk density of the full peat profile was initially around 40 kg m-3, increasing to 50 kg m-3 as the peat layers grew older. Some studies (Clymo, 1991; Novak et al., 2008) noted a decline in bulk density with depth due to compaction. However, the simulated peat column does not exhibit such a decline with depth, instead being highly variable down the profile as found in other studies (Tomlinson, 2005; Baird et al., 2016). Freezing of the lower layers inhibited decomposition, with the result that bulk densities remained higher relative to other partially frozen or unfrozen layers. The pore space and permeability are linked to the compaction of peat layers. Therefore, when the peat bulk density increased, pore space declined from 0.95 to 0.93 reducing the total permeability of peat layers that in turn reduced the amount of percolated water from the peat layers to the mineral soil.

l. 421: Are there any vegetation reconstructions available for these sites? Pollen cores that can help determine if the model successional sequence is reasonable?

Response: For majority of its peatland history, different species of mosses occupied the Stordalen mire. The model predicted correctly the dominance of wet PFT during 4000-3000 cal. BP. However, there was a certain period between 700-1700 cal. BP when graminoids were again the dominant PFT, but we could not reproduce that period due to the climate forcing used here, as explained in the text. See lines: 527-533 in the

RM.

Revised text: Mosses emerged as the dominant PFT at the beginning of the simulation, while 300-400 years after peat inception shrubs started establishing in the higher elevated patches as a result of a lowering of WTP. Graminoids were not productive during the entire simulation period apart from the period 4-3kyr cal. BP (Kokfelt et al., 2010). The model predicted correctly the dominance of graminoids, characteristic of wet conditions, during 4-3kyr cal. BP. However, a period of graminoid dominance between 700-1700 cal. BP was not accurately captured. One explanation can be the absence of decadal and centennial climate variability in the adopted climate forcing data, resulting in an "averaging out" of moisture status over time that elminates wet episodes needed for graminoids to be sufficiently competitive.

Fig 1: why are the mosses all different colours? Can this diagram be simplified - like only a couple grass instead of that dark mat? Should permafrost maybe be 'frozen soil' or maybe distinguish seasonally frozen soil from perenially frozen? Why is the permafrost bubble circular? Would the model really have a different bottom permafrost depth between its tiles in the same gridcell? I can understand a different top depth but not really a bottom.

Response: Our moss colours are different because they depict different stages of the moss growth cycle. However, for simplicity we have changed it to single colour. Graminoids numbers are also reduced. We have changed the text from permafrost to frozen soil and removed the circularity (see Fig. 1 in the RM). In principle, our model can have different ALD values in each patch based on the soil temperature and soil water content in that patch. Wet patches can have greater ALD than dry patches (see Fig. 9 in the RM).

Fig 2: Perhaps choose a different acronym than UM since that is also used in the MS to talk about a model.

Response: We have changed it to UMS (Upper mineral soil). See Fig. 2 in the RM.

[Figure]

Revised caption: Fig. 2 in the RM - Root fractions in the upper (UMS) and lower mineral soil (LMS) layers as a function of peat depth (m). The broken lines represent root fractions in UMS and solid lines indicate fractions in the LMS.

Fig. 6 I find the acronym choice non-sensible. Why does the final S of deciduous shrubs be S and not a D? Not a big deal but it makes it harder to quickly remember what the acronym stands for. Response: We used HSS in the paper since it is the acronym for High Summergreen Shrubs and LSS for Low Summergreen Shrubs (see Table 1 in the RM). These are the most common acronyms used in LPJ-GUESS publications (Wolf et al., 2008; Miller and Smith, 2012). We have revised the figure (see Fig. 6 in the RM).

Fig 7: No description of the X and Z in the caption. What do Top, Middle, and Bottom really correspond to? This gets back to my earlier comment that I don't understand how your soil layers were divided.

Response: We have now explained X and Z in the caption. There are 4739 peat layers and they were aggregated in to number of sublayers for the soil temperature calculation. We start with three sublayers of equal depth and add a new sublayer for every half a meter peat depth increment. We adopted this scheme for soil temperature because over the time these individual layers become so thin and numerous that they slow down the numerical soil temperature calculations. In total, seven sublayers formed at the Stordalen site and in figure the three sublayers are shown as top (average of layers 6+7), middle (average of layers 3+4+5) and bottom (average of layers 1+2). See lines: 172-181 in the RM. Revised text: To simulate permafrost, peat layer decomposition and cycles of freezing and thawing, the soil temperature at different depths must be calculated correctly. In the Arctic version of LPJ-GUESS as described by Miller and Smith (2012), mineral soil layers (i.e. below the peat layers added in this study) are subdivided into 20 sublayers of 10 cm thickness to calculate soil temperature at different depths. In our implementation, new peat layers are added on top of these mineral soil layers. To overcome computational constraints for millennial simulations we aggregate

the properties of the individual annual peat layers into thicker sublayers for the peat temperature calculations, beginning with three sublayers of equal depth and adding a new sublayer to the top of previous sublayers after every 0.5 m of peat accumulation. This resulted, for example, in seven aggregate sublayers for the Stordalen simulations described in Section 2.4. The result is a soil column with a dynamic number of peat sublayers, 20 mineral soil layers and multiple "padding" layers to a depth of 48 m. A single layer of snow is included, as in existing versions of the model.

Revised Caption: Fig. 7. (a) Total simulated peat ice fraction (10-year moving average) over 4700 years at Stordalen. Peat layers corresponding to annual litter cohorts were aggregated to top (top 1 m), middle (middle 1 m) and bottom (lower 1.5 m) for display. (b) Total simulated ice fraction for 1900-2100 following the RCP8.5 scenario (see Fig. A6 for the RCP2.6 scenario results), (c) Total simulated mean September active layer depth for the last 4700 years and (d) for 1900-2100 at Stordalen following the RCP8.5 scenario (FTPC8.5) and RCP2.6 scenario (FTPC2.6).

Fig 8: As I said in the general comments, this figure does not give much confidence when combined with the NEE results.

Response: As mentioned earlier, NEE outputs for the other three sites are almost within the range of observed NEE values (see Table 5 in the RM), albeit with some differences. The recent short-term NEE values are not the right criteria to judge whether the model is doing the right job or not because they vary a lot spatially as well as temporally and since the peatland landscape is such a heterogeneous site, the NEE values vary between each points. Though, large-scale fluxes can be obtained from eddy flux tower but they also showed high variability (Lafleur et al., 2003; Aslan-Sungur et al., 2016). In this study, some sites (Fajemyr and Degero Stormyr) are relatively disturbed sites with high N deposition which might have influenced their NEE fluxes (Lund et al., 2007). The other factor of large uncertainty in NEE in Fajemyr is non-inclusion of trees. Also, water borne carbon fluxes (DOC) and $CH_4$ are not yet considered in our model (but are under development; e.g. Tang et al., 2015b). Inclusion of these factors would minimize

the uncertainty. This is the reason we didn't do any future predictions for these sites. However, comparatively less disturbed sites showed reasonable simulated NEE values (see Fig.11 in the RM). We believe, the right evaluation for the peat carbon balance can be extracted from long-term carbon accumulation values (LARCA) and we find a close match between modelled and observed LARCA values. In our companion paper, we have found the model is able to capture the right LARCA value across many regions. See lines: 540-548 in the RM.

In Fig. 11 of the RM, we presented (a) annual simulated NEE (kg C m-2 yr-1) for Stordalen and (b) relationship between observed and modeled annual NEE (kg C m-2 yr-1) for three Scandinavian peatland ecosystems (Table 5; observed NEE data from (Aurela et al., 2007; Lund et al., 2007; Sagerfors et al., 2008; Aslan-Sungur et al., 2016)). EC = eddy covariance (flux tower) data; CH = chamber flux measurements.

Revised text: For the additional evaluation sites, modelled dominant vegetation cover, LARCA and WTP were in good agreement with the observed values for the three selected sites at which this information was available. Under the present climate, Stordalen was simulated to be a small sink for atmospheric CO2, in agreement with observed NEE (see Fig 11). NEE interannual range is likewise close to observations for the other Scandinavian sites (Table 5). However it is uncertain whether recent annual observations of NEE necessarily reflect the long-term peatland carbon balance, in view of high variability on multiple timescales. For example, Fajemyr has switched between source (14.3-21.4 g C m-2 yr-1 in 2005-2006; 23.6 g C m-2 yr-1 in 2008) and sink (-29.4 g C m-2 yr-1 in 2007; -28.9 g C m-2 yr-1 in 2009) conditions in recent years, and this variability has been attributed to disturbances and intermittent drought conditions (Lund et al., 2012).

Also, see Table 5 in RM where observed dominant vegetation cover, long-term apparent rate of carbon accumulation (LARCA), short-term net ecosystem exchange (NEE), and annual water table position (WTP) compared with mean modelled values (1990-2000) for the 3 grid points in Scandinavian region were included

Fig A1 - perhaps add total water (liquid and frozen) so we can see if the total content was changing and it wasn't just changing phase.

Response: We have added a new panel showing the total water and ice in cm (see Fig. A4 in the RM). In the figure, total water is the melted water and total ice is the frozen water.

Table 2: density is needing the o as an subscript. Also please bring these all into the main text, it is annoying to have to search out the table when one is reading the text (and it is often not mentioned that one needs to search for a table...)

Response: We have added a subscript to the density parameter. We have also included all the parameter values in the text and we also now refer to Table 2 in RM when a new constant is first mentioned See lines: 150-153, 156-158, 166-168, 219, 222, 231 and 234 and Table 2 in the RM.

Table 5: WTP units? Please put in proportions of the veg so we can tell if the proportions modelled are in any way correct rather than just presence/absence.

Response: The WTP unit is cm and we have included in the text (see Table 5 in the RM). However, we unfortunately couldn't find total vegetation proportion data for these sites.

[revised manuscript text omitted]

---

## Author Comment (AC4) · 21 Mar 2017

We appreciate the time and effort spent by the reviewers in reviewing this manuscript. We have addressed all the issues indicated in the review reports and believe that the revised version will meet the journal's publication requirements.

Substantive comments

Model choice and model scale

The authors note the following:

"Model formulations of peat accumulation and decay have been proposed and demonstrated at the site scale (Frolking et al., 2010) but have not yet, to our knowledge, been implemented within the framework of a DGVM, or applied at larger spatial scales than a single study site or landscape."

The authors are right, but they then go on to apply their landscape-scale model (or land surface scheme) to individual sites, so we do not get to see what the LPJ-GUESS model does at larger scales in comparison to a series of smaller site models. The authors also provide a very limited review of other peatland models. At least two other models have been developed – MILLENNIA (Heinemeyer et al., 2010) and DigiBog (e.g., Morris et al. (2012) and Morris et al. (2015)) – and it might be useful to acknowledge what these models are capable of doing and their limitations.

Response: Our model can be employed at the site-scale and, where climate forcing is available at a sufficient resolution, at the regional scale. We focused on site-scale runs in this study because we wanted to describe the model processes and their evaluation using data from well-studies sites such as Stordalen and Mer Bleue. However, in work that was completed in the time since this paper was submitted, we have run the model for 180 sites evenly spread across the pan-Arctic and shown that the model can produce reasonable predictions of past and present carbon accumulation rates at regional scale. See our companion paper in discussion - Biogeosciences Discuss., doi:10.5194/bg-2017-34, 2017.

We have now expanded our acknowledgements of the work done by other groups and referred to them in relevant places. See lines: 35-45 in the revised manuscript (RM). We compared the functionality and scope of a representative set of current peatland models in Table S1 in the RM (there are many other models apart from the one mentioned in the Table S1) but this list we think is not suitable for the paper. Could be included in the appendix though.

Revised text- Dynamic global vegetation models (DGVMs) are used to study past, present and future vegetation patterns from regional to global scales, together with

associated biogeochemical cycles and climate feedbacks, in particular through the carbon cycle (Smith et al., 2001; Friedlingstein et al., 2006; Sitch et al., 2008; Strandberg et al., 2014; Zhang et al., 2014). Only a few DGVMs include representations of the unique vegetation, biophysical and biogeochemical characteristics of peatland ecosystems (Wania et al., 2009a, b; Kleinen et al., 2012; Tang et al., 2015). Model formulations of multiple peat layer accumulation and decay have been proposed and demonstrated at the site scale (Frolking et al., 2010; Heinemeyer et al., 2010) but have not yet, to our knowledge, been implemented within the framework of a DGVM. However, peatland processes are included in some other types of model frameworks (Morris et al., 2012; Alexandrov et al., 2016; Wu et al., 2016) and been shown to perform reasonably for peatland sites. Large area simulations of regional peatland dynamics have been performed by (Kleinen et al., 2012; Schuldt et al., 2013; Stocker et al., 2014; Alexandrov et al., 2016) (see Table S1 in the RM).

Table S1 in the RM shows comparison of functionality and scope of a representative set of current peatland models.

The authors note that vegetation in their modelled domain can develop into patches and that each patch is represented by a different soil column. The authors seem to suggest that patches can emerge over time, but, if that is so, how can a different soil column be assigned a priori to each patch? The authors also suggest that water can flow between patches, which makes sense, but do not indicate how such flows are simulated (see point 3 below).

Response: We are sorry of this is a little unclear. The number of patches in our model is fixed at the outset. Each patch has its own soil column (composed of mineral and, eventually, peat layers) and dynamic vegetation properties. Vegetation within the patches competes for water and sunlight but there is no competition or communication between patches except for the distribution of water. Our model randomly distributes the carbon in the start of the simulation over the static mineral soil layers leading to an initially heterogeneous surface (different patch heights). As they accumulate C, these individual

patches develop their own hydrologies and water holding capacities leading to different patch water heights. At the end of each day of the simulation, we take the mean of water table position (WTP) across all patches, and this is referred to as mean landscape WTP in the manuscript. The water flow from higher patches to lower patches is based on mean landscape WTP. For instance, hollows have lower peat C mass leading to lower water holding capacity overall and a lower water height relative to hummocks. We add or remove the amount of water required to match the mean landscape WTP in each patch, in each time step (see below for a more detailed description).

Model complexity and process and parameter redundancy

LPJ-GUESS is a complicated model – it does many things. In choosing what processes to represent in a model it is important to consider process and parameter redundancy. For example, it may seem intuitively correct to include all obvious plant functional types, but the inclusion of some may add little to the predictive power of the model. For example, how does the model behave if litter production is confined to, for example, a single shrub PFT; do the model's results change substantially? I wonder too whether the litter production functions in the model could be replaced with a simpler function and the model results remain essentially the same? I am not suggesting the authors change the model and re-run it. It would, however, be useful to see brief consideration of why the model has been set up as it has been. Currently, the model set up is described rather than justified. An important paper on this topic is that by Crout et al. (2009) who show, for example, that a well-established and popular wetland CH4 model is over-complicated and can achieve the same predictive success in much simpler form. Models are often more complicated that they need to be.

Crout NMJ, Tarsitano D, Wood AT. 2009. Is my model too complex? Evaluating model formulation using model reduction. Environmental Modelling and Software 24: 1–7, doi: 10.1016/j.envsoft.2008.06.004.

Response: This is a very good point. In fact, we were forced to make decisions to

balance complexity and utility while developing our model. For example, we initially chose four PFTs (mosses, two dwarf shrubs and graminoids) and found that though the model was performing fairly well for the Stordalen site, it performed less than satisfactorily when we applied the model to temperate sites which have higher plant diversity than the Stordalen subarctic mire. Further investigation revealed the litter carbon mass deposited by the four PFTs was not sufficient (less than the reported values) leading to shallow peat heights in temperate regions. Therefore, we decided to include high summergreen shrubs (HSS) in the model, which is one of the more important PFTs in temperate peatland ecosystems (Moore et al., 2002). HSS establishes when the growing degree days (GDD) is higher than 1000 degree-days, thereby limiting HSS establishment in colder regions. However, adding high evergreen shrubs (HSE) did not substantially improve the predictive power of the model so we excluded it from the set up.

A further example is the treatment of soil temperature in the model. There are thousands of peat layers in the later stages of our simulations, and one approach to calculating layer temperatures for use in the decomposition equation would be to use a finite-difference numerical scheme considering all these layers in each step. It is questionable if such detail is warranted however, and it would be difficult to evaluate such a profile, so we opted for a scheme in which we aggregated the peat layers to a smaller number of layers for use in the numerical scheme, with the exact number increasing from 3 to 7 as the peat depth increases. This method was sufficient to model the active layer depth seasonally and annually.

Hydrological components of the model

I found the explanation of the hydrological part of the model difficult to follow. In particular, it was unclear how the model predicts the soil moisture content of the peat above the water table. The authors note that rates of peat decomposition depend on peat wetness and suggest that the highest rates of decay occur when the peat is at field capacity, but they do not say how they modelled soil moisture content (as opposed to

water-table position). Equation 7 is a balance equation that shows the different inflows into, and outflows from, the model. However, I could not find any discussion of how water inputs are allocated separately between the unsaturated and saturated zones.

Response: We use a simple bucket scheme when adding water (rain or snowmelt) from the current WTP to the top of the peat column formed by individual peat layers giving a new WTP in each time step. In our model peat layers above the WTP are thus assumed to be completely unsaturated. We simulate water and ice in each peat layer of each individual patch and convert them into water and ice content by dividing the amount of ice and water with total water holding capacity. If layer is totally frozen (100% ice), then it cannot hold additional water. In partially frozen soil, the sum of the fractions of water and ice is limited to water holding capacity of that layer. The soil water content determines the peat decomposition rate in individual layers.

The authors are also unclear on how lateral flows of water occur in the model. On lines 254-256 they note:

"We equalize the WTP of individual patches according to the mean WTP of the landscape. The higher patches loses water if the WTP is above the mean WTP of the landscape while the lower patches receive water."

This description is too general and it is not clear numerically how water is moved across the landscape. I assume the model has lateral boundary conditions but such conditions are not mentioned in the paper. These can have a profound effect on how the model functions hydrologically so should be discussed and justified.

Response: We calculate the landscape WTP (as discussed above) and add and remove the amount of water from each patch required to match the landscape WTP. See below the representation how it is done.

MWTP= $\sum PWTP\_i/n$

where MWTP is the mean WTP across all the patches, PWTPi is the water table position in individual patches (i) and n is the total number of patches. The water to be added to or removed from each patch with respect to mean WTP (MWTP) in each patch, i.e. lateral flow (LF) is given by:

DWTPi = PWTP_i - MWTP

LFi = DWTP . $\Phi$a

where DWTPi is the difference in the patch (i) and MWTP and LFi is the total water to be added or removed with respect to MWTP in each patch (i). If the WTP is below the surface then the total water is calculated by the difference in WTP (water heights) multiplied by average porosity ($\Phi$a). When the WTP is above the surface then $\Phi$a is not included in the calculation. This exchange of water between patches is implemented after the daily water balance calculation.

There seems to be some confusion too in how different processes are reported. For example, 'R' is defined as surface runoff in Equation 7 but later (in Equation 9) is described as a function of base runoff which seems to be some type of subsurface flow.

Response: R in Eq. 7 is the total runoff – base runoff plus and surface runoff. We have corrected it in the text. See lines: 201-203 in the RM. We have made the changes and termed the base flow as BR.

Revised text: where W is the total water input, P is the precipitation, ET is the evapo-transpiration rate, R is the total runoff, DR for the vertical drainage and LF (see section 2.1.7 below) is the lateral flow within the landscape depending upon the relative position of the patch.

I recommend section 2.1.4 is re-written to make it clearer and that it is accompanied by a new diagram which shows all of the components of the hydrological budget as represented in the model (the current Figure 1 is not sufficient for this purpose).

Response: We have re-written section 2.1.4 and 2.1.4 (see lines: 191-236 and 263-289

in the RM)

4. Representation of Stordalen and of soil ice

The authors compare their simulation of the Stordalen mire to a reconstruction by Kokfelt et al. (2010), a paper which I have not read. I think it would be useful if the authors indicated in more detail how Kokfelt et al. estimated past peat thicknesses of the mire. More fundamentally, I am not clear on the appropriateness of considering peat thickness from one location at a site. My understanding is that Stordalen is a palsa mire in which case it will comprise elevated palsas – large ombrotrophic hummocks – formed by the growth of ice lenses, and intervening minerotrophic areas that form after wastage and collapse of the ice lenses. The authors note on line 543 that their model cannot simulate peat subsidence due to permafrost thaw. What is not clear is whether it can also simulate the palsa cycles that would have occurred prior to the recent warming of the climate in the region. As far as I can tell the model is not capable of simulating ice lenses.

Response: We have included a short description of how Kokfelt et al. 2010 estimated past peat thickness of the Stordalen mire. They used radioisotope dating at several depths and a Bayesian modeling technique to reconstruct the thickness of the mire. They have also used peat cores from nearby lakes to reconstruct the past climate influence on vegetation dynamics, hydrological changes and nutrient flow within the catchment. We discussed this in lines: 313-315 in the RM.

Response: We agree that the ideal case is to compare the model with multiple peat cores from the same site this is not feasible in this case because this data is not available for the Stordalen site. Though the model has peatland and permafrost functionality, it doesn't yet simulate ice lenses, palsas and palsa expansion and contraction cycles. In the future modifications, we may include these features.

Revised Text:

Based on radioisotope dating of peatland and lake sequences supplemented with Bayesian modelling, Kokfelt et al. (2010) inferred that the peat initiation started ca. 4700 calendar years before present (cal. BP) in the northern part and ca. 6000 cal. BP in the southern part.

Figure 4 shows the 'observed' peat thickness (the reconstructed peat thickness) at different times during Stordalen's development and the modelled thickness. The authors provide a 95% CI around the 'observed' values but say the CI was inferred from the model runs. Did the model actually produce multiple peat thicknesses for different patches, in which case why don't the authors show the spread of outputs from the model?

Response: Yes, the 95% confidence interval is calculated from the individual peat depths simulated for each of the modelled patches. The model simulated 10 different peat thickness trajectories, one for each patch (see Fig. 4 in the RM). We originally thought that showing the spread would not add much to the figure so we only included 95% confidence interval. However, we have now updated the Figure and its caption to remove this source of uncertainty.

Finally, a more minor issue, but one that is important to address, is that it is not always clear what units are used in different parts of the model. They are given in some places but not others – I recommend that whenever a parameter or variable is first defined its units are given.

Response: We have now gone through the paper and made the required changes to include the units whenever a parameter or variable is first introduced. See lines: 150-153, 156-158, 166-168, 219, 222, 231 and 234 and Table 2 in the RM.

General Comments

However, current DGVMs lack functionality for the representation of peatlands, an important store of carbon at high latitudes

Comment: And also in parts of the tropics.

Response: We have revised the sentence (see lines: 10-13 in the RM).

Revised text: Dynamic global vegetation models (DGVMs) are designed for the study of past, present and future vegetation patterns together with associated biogeochemical cycles and climate feedbacks. However, most DGVMs do not yet have detailed representations of permafrost and non-permafrost peatlands, which are an important store of carbon particularly at high latitudes.

Our approach employs a dynamic multi-layer soil with representation of freeze-thaw processes and litter inputs from a dynamically-varying mixture of the main peatland plant functional types; mosses, dwarf shrubs and graminoids.

Comment: I recommend a colon here (see line: 16 in the RM).

Response: We have revised the sentence.

Revised text: Our approach employs a dynamic multi-layer soil with representation of freeze-thaw processes and litter inputs from a dynamically-varying mixture of the main peatland plant functional types: mosses, shrubs and graminoids.

We found that the Stordalen mire may be expected to sequester more carbon in the first half of the 21st century due to milder and wetter climate conditions, a longer growing season, and $CO_2$ fertilization effect, turning into a carbon source after mid-century because of higher decomposition rates in response to warming soils.

Comment: "and *a* CO2" (add 'a')

Response: We have revised the sentence (see line 23 in the RM).

Revised text: We found that the Stordalen mire may be expected to sequester more carbon in the first half of the 21st century due to milder and wetter climate conditions, a longer growing season, and the $CO_2$ fertilization effect, turning into a carbon source after mid-century because of higher decomposition rates in response to warming soils.

[Figure]

Peatlands are a conspicuous feature of northern latitude landscapes (Yu et al., 2010), of key importance for regional and global carbon balance and potential responses to global change.

Comment: A bit vague. Change of what? I assume climate is meant. I suggest re-wording to be more specific.

Response: Yes, we forgot to add "climate". We have now revised the sentence (see line: 26-27 in the RM).

Revised text: Peatlands are a conspicuous feature of northern latitude landscapes (Yu et al., 2010), of key importance for regional and global carbon balance and potential responses to global climate change.

In the past 5-10 thousand years they have sequestered approximately 200-550 Pg C across an area of approximately 3.5 million km2 (Gorham, 1991; Turunen et al., 2002; Yu, 2012).

Comment: Give as a number rather than a mix of numbers and words? The higher end is more likely.

Response: We have revised the sentence (lines: 27-28 in the RM)

Revised text: In the past 10,000 years (10 kyr) they have sequestered 550 $\pm$100 PgC across an area of approximately 3.5 million km2 (Gorham, 1991; Turunen et al., 2002; Yu, 2012).

Peatlands are also considered one of the major natural sources of methane, contributing significantly to the greenhouse effect (IPCC, 2013; Lai, 2009; Whiting and Chanton, 1993)

Comment: Considered or actually are one of the main sources?

Response: We have revised the sentence (see lines: 29-30 in the RM).

Revised text: Peatlands are one of the major natural sources of methane, contributing significantly to the greenhouse effect (Whiting and Chanton, 1993; Lai, 2009; IPCC, 2013).

The majority of northern peatland areas coincide with low altitude permafrost (Wania et al., 2009a). Comment: Really? The majority?

Response: We have revised the sentence (see lines: 30-31 in the RM).

Revised sentence: Around 19% (3556 × 103 km2) of the soil area of the northern peatlands coincides with low altitude permafrost (Tarnocai et al., 2009; Wania et al., 2009a).

There is a scientific consensus that the climate is likely to warm in the coming century, and that the warming will be amplified in northern latitudes, relative to the global mean trend (IPCC, 2013).

Comment: The present century? In which case the climate has already warmed and is predicted to continue doing so.

Response: Agreed. We have revised the sentence (see lines: 46-47 in the RM).

Revised text: Climate warming is amplified in northern latitudes, relative to the global mean trend, due to associated carbon-climate feedbacks (IPCC, 2013).

Uniquely among existing large-scale (regional-global) models, we thus account for feedbacks associated with hydrology, peat properties and vegetation dynamics, providing a basis for understanding how these feedbacks affect peat growth on the relevant centennial-millennial time-scales and in different climatic situations.

Comment: Okay, but you actually apply your model at the site scale, so your implementation is not really different from an implementation of the Holocene Peat Model for example. Response: We have explained this part above.

Five PFTs characteristic of peatlands – mosses (M), graminoids (Gr), deciduous and

evergreen low shrubs (LSS and LSE) and deciduous high shrubs (HSS) – are included in the present study.

Comment: Why were five chosen? Why not three PFTs, or 12?

Response: We have addressed this point in an early response to a question by the reviewer.

A one-dimensional soil column is represented for each patch (defined below), divided vertically into four distinct layers: a snow layer of variable thickness, a litter/peat layer of variable thickness, a mineral soil column with a fixed depth of 2 m (with further sublayers of thickness 0.1 m), and finally a "padding" column of m depth (with thicker sublayers) allowing to simulate accurate arctic soil thermal dynamics (Wania et al., 2009a). The insulation effects of snow, phase changes in soil water, precipitation and snowmelt input and air temperature forcing are important determinants of daily soil temperature dynamics at different depths.

Comment: Can the physical properties (e.g., porosity, hydraulic conductivity) of this layer vary with depth?

Response: Porosity is a function of bulk density, and influenced by total mass remaining in each peat layer. If the layers are highly decomposed their bulk density increases and porosity will decline. We do not consider the hydraulic conductivity explicitly in this study, but the drainage is affected by the permeability of peat layers and the saturation limit of the mineral soil underneath.

Fresh litter debris decomposes through surface forcing until the last day of the year.

Comment: It's not clear what this means.

Response: When the litter (leaves and stems, where appropriate) is dropped on the ground surface it doesn't become a part of peat column (formed of multiple layers – see above) instantaneously. This litter then decomposes at rates depending on the surface conditions in that year, i.e. surface temperature and moisture, becoming the

top layer in the peat layer of the soil column the following year. So, in our framework, we decompose the litter mass present on the peat surface for the first year before it transforms into a peat layer. However, for dead roots, we add them directly to the peat layers where they belong (see lines: 106-110 in the RM).

Revised text: Fresh litter debris decomposes on the surface through exposure to surface temperature and moisture conditions until the last day of the year. The decomposed litter carbon is assumed to be released as respiration directly to the atmosphere while any remaining litter mass is treated as a new individual peat layer from the first day of the following year, which then underlies the newly accumulating litter mass.

This layer can be composed of up to 17 carbon components (g C m-2), namely leaf, root, stem and seeds from shrubs, mosses and graminoids (see Table 1) and the model keeps a track of these layer components as they decompose through time.

Comment: That's a lot of components. Does the model need to be this complicated or could (should) it be more parsimonious? Is it over-parameterised?

Response: We believe that this distinction is important because each litter component plays an important part in peat formation and the quantity and quality of litter is also different for each PFT component. For example, stem wood decomposes at a much slower rate than other components of shrubs, while root turnover directly enter subsurface peat layers where they belong.

Total peat depth is derived from the dynamic bulk density values calculated for individual peat layers.

Comment: I'm confused. How many peat layers are there? Just two - acrotelm and catotelm - or one for each year of the model simulation?

Response: We appreciate this ambiguity now, spotted by both reviewers. It's the latter – one for each year of the simulation. For Stordalen, 4739 + 100 peat layers were simulated, i.e. one peat layer for each of the 4739 years after inception until year 2000,

followed by a 100-year projection from 2001 to 2100. For Mer Bleue, it was 8400 + 100 layers. We cannot show that many layers in a figure so we simplified the representation in Fig. 1 in the RM (see lines: 91-98 in the RM).

Revised text: A one-dimensional soil column is represented for each patch (defined below), divided vertically into four distinct layers: a snow layer of variable thickness, one dynamic litter/peat layer of variable thickness corresponding to each simulation year (e.g. 4739 + 100 layers by the end of the simulations, described in Section 2.4 below, for Stordalen), a mineral soil column with a fixed depth of 2 m consisting of two sublayers: an upper mineral soil sublayer (0.5 m) and a lower mineral soil sublayer (1.5 m), and finally a "padding" column of 48 m depth (with 10 sublayers) allowing the simulation of accurate soil thermal dynamics (Wania et al., 2009a). The insulation effects of snow, phase changes in soil water, precipitation and snowmelt input and air temperature forcing are important determinants of daily soil temperature dynamics at different depths.

Comment: Why is this term given thus and not as single number? Response: Yes, we have revised the equation (see Eq. 4 in the RM).

The acrotelm is the top layer in which water table fluctuates leading to both aerated and anoxic conditions.

In our implementation, new peat layers are added on top of these mineral soil layers. To overcome computational constraints for millennial simulations we aggregate the properties of the individual annual peat layers into thicker sublayers for the peat temperature calculations, beginning with three sublayers of equal depth and adding a new sublayer to the top of previous sublayers after every 0.5 m of peat accumulation.

Comment: Some recent papers suggest the distinction between acrotelm and catotlem is not helpful. See, e.g., Morris et al. (2011) Ecohydrology 4, 1-11.

Comment: Okay; so there are multiple peat layers. This could have been made clearer

above.

Response: We have explained this above (see lines: 91-98 in the RM).

DR for the drainage

Comment: Should this be defined here as vertical drainage? Response: We have revised it to vertical drainage (see line 202 in the RM).

Revised text: where W is the total water input, P is the precipitation, ET is the evapo-transpiration rate, R is the total runoff, DR for the vertical drainage and LF (see section 2.1.7 below) is the lateral flow within the landscape depending upon the relative position of the patch.

R=BR

Comment: Why the italics here and not elsewhere?

Response: We have removed the italics. Thanks.

Loss of the water through drainage/percolation depends on the permeability of peat layers and the saturation limit of the mineral soil underneath.

Comment: Only vertical drainage seems to be simulated. In many (most) ombrotrophic peatlands, drainage is predominately a lateral process - the peatland drains to its margins. Is lateral drainage accounted for in the model? If so, what relationship is used? What are the dimensions/units of permeability? Do the authors mean intrinsic permeability or hydraulic conductivity?

Response: We have not included an explicit description of the lateral drainage but our runoff function, R, implicitly takes into account the lateral drainage, and it is also dealt with through our lateral distribution of water among patches. We mean intrinsic permeability (0-1), which is calculated based on peat bulk density (kg m-3 ; see Eq. 11 in the RM).

[Figure]

become highly compressed under accumulating peat mass and humified by anoxic decomposition (Clymo, 1991).

Comment: But you note earlier that dry bulk density often does not show depth dependency in the 'catotelm'.

Response: Simulating bulk density is a challenge. In some peatlands, it may increases with depth due to compaction (Clymo, 1991) but other studies have shown no net increase in the bulk density with depth in some other locations (Baird et al., 2016). In our study, the simulated bulk density is a function of the total mass remaining and in the peat profile it varies between 40-102 kg m-3 for Stordalen. Ryden et al. (1980) given a range of 45-230 kg m-3 (see page 41 and Table 5 and 6 in their paper) and our values are well within this range. We also find bulk density doesn't decline with depth in our profile. Since the lower layers were frozen, they didn't decompose significantly and their bulk densities remain higher relative to other partially frozen or unfrozen layers. The value referred to by the reviewer is the mean value of the entire simulated peat profile and it was lower than 50 kg m-3 since the majority of peat layers are not highly compacted as a result limited decomposition due to permafrost or high water contents (see lines: 429-438 in the RM)

Revised text: When the peat layers had decomposed sufficiently and lost more than 70% of their original mass (Mo), their bulk density increased markedly. The observed monthly and annual WTP for the semi-wet patches and mean annual ALD were very near to the simulated values (see Figs. 8, 9 and A5). The simulated bulk density varies between 40-102 kg m-3 and the mean annual bulk density of the full peat profile was initially around 40 kg m-3, increasing to 50 kg m-3 as the peat layers grew older. Some studies (Clymo, 1991; Novak et al., 2008) noted a decline in bulk density with depth due to compaction. However, the simulated peat column does not exhibit such a decline with depth, instead being highly variable down the profile as found in other studies (Tomlinson, 2005; Baird et al., 2016). Freezing of the lower layers inhibited decomposition, with the result that bulk densities remained higher relative to other

partially frozen or unfrozen layers. The pore space and permeability are linked to the compaction of peat layers.

The amount of water draining from the peat column to the mineral soil is calculated by integrating permeability across all the peat layers (i)

Comment: Not clear what is meant by integration here. If simulating vertical drainage, then perhaps it would make sense to use a harmonic mean.

Response: We have revised the sentence (see lines: 228-229 in the RM).

Revised text: The amount of water draining from the peat column to the mineral soil is calculated by integrating permeability across all the peat layers (i).

Change of porosity ($\Phi$) due to compaction is captured by a relationship to bulk density:

Comment: I assume this should be 'drainable porosity' which is not the same as total porosity. How is the moisture content of the peat above the water table simulated?

Response: Yes, it is a drainable porosity. We have not calculated moisture content above the water table – see the response to the reviewer's earlier comment.

Shrubs are vulnerable to waterlogged and anoxic conditions (Malmer et al., 2005) and establish only when annual WTP deeper than -25 cm below the surface.

Comment: Better to say 'relative to'?. A negative value below the surface means something above the surface. A negative depth means a positive value (something above the surface). This sentence would be simpler if you just say it was 25 cm below the surface.

Response: We agree, and have revised the text (see lines 254-255 in the RM).

Revised text: Shrubs are vulnerable to waterlogged and anoxic conditions (Malmer et al., 2005) and establish only when annual WTP is deeper than 25 cm relative to the surface.

The model is initialised with a random surface represented by uneven heights of individual patches (10 in the simulations performed here).

Comment: Okay, but do non-random patterns subsequently form in the model?

Response: Yes, we find that when we start the model with a flat surface we get heterogeneous patch/peat heights and vegetation composition after several years (see Fig. 1 in this document).

Water is redistributed from the higher elevated sites to low depressions through lateral flow (see Eq. 7).

Comment: But equation 7 is a water-balance equation. It does not indicate how LF is calculated

Response: We have added the lateral flow equations in section 2.1.7 (see our reply to an earlier comment above and section 2.1.7 in the RM).

We equalize the WTP of individual patches according to the mean WTP of the landscape. The higher patches loses water if the WTP is above the mean WTP of the landscape while the lower patches receive water.

Comment: Okay, but how does this equalisation process work?

Response: We have revised the sections 2.1.5 and 2.1.7 (see our reply to an earlier comment above)

Permafrost underlying elevated areas have been degraded as a result of climate warming in recent decades, with an increase in wet depressions modifying the overall carbon sink capacity of the mire (Christensen et al., 2004; Johansson et al., 2006; Malmer et al., 2005).

Comments: Replace with 'has'. For more recent work see Swindles et al. (2015) Scientific Reports 5, 17951.

Response: We have revised the text and added the reference (see lines: 307 in the RM).

Revised text: Permafrost underlying elevated areas has been degraded as a result of climate warming in recent decades, with an increase in wet depressions modifying the overall carbon sink capacity of the mire (Christensen et al., 2004; Malmer et al., 2005; Johansson et al., 2006; Swindles et al., 2015).

To evaluate the generality of the model for regional (e.g. pan-Arctic) applications, we validated its performance against observations and measurements at Mer Bleue (45.40° N, 75.50° W, elevation 65 m a.s.l.), a raised temperate ombrotrophic bog located around 10 km east of Ottawa, Ontario (Fig. 3). Comment: Mer Bleue is a long way from the Arctic - as you note, it is a temperate mire. Response: We have revised the sentence and removed the word pan-Arctic (see lines: 319-321 in the RM).

Revised text: To evaluate the generality of the model for regional applications, we compared its predictions to observations and measurements at Mer Bleue (45.40° N, 75.50° W, elevation 65 m a.s.l.), a raised temperate ombrotrophic bog located around 10 km east of Ottawa, Ontario (Fig. 3).

This bog is mostly covered with Sphagnum mosses (S. capillifolium, S. magellanicum) and also dominated by a mixture of evergreen (Chamaedaphne calyculata, Ledum groenlandicum, Kalmia angustifolia) and deciduous shrubs (Vaccinium myrtilloides).

Comment: This is an out of date name. It is now Rhododendron groenlandicum (Oeder) Kron. Response: Thanks, we have renamed it (see line: 325 in the RM).

Revised text: The bog surface is characterized by hummock and hollow topography. This bog is mostly covered with Sphagnum mosses (S. capillifolium, S. magellanicum) and also dominated by a mixture of evergreen (Chamaedaphne calyculata, Rhododendron groenlandicum, Kalmia angustifolia) and deciduous shrubs (Vaccinium myrtilloides).

In the standard (STD) experiment, a total of 94.96 kg C m-2 of peat was accumulated over 4700 years, leading to a cumulative peat depth profile of 2.11 m predicted for the present day Comment: Just one depth? Would not multiple depths have been predicted, one for each vegetation patch? See my referee's report.

Response: We have given a range in Table 4 and included a new figure showing different peat trajectories (Fig. 4 in the RM). This is the range 1.9 - 2.2 m (see lines: 406-408 in the RM).

Revised text: In the standard (STD) experiment, a total of 94.6 kg C m-2 (91.4-98.9 kg C m-2) of peat was accumulated over 4700 years, leading to a cumulative peat depth profile of 2.1 m (1.9-2.2 m) predicted for the present day (Fig. 4), comparable to the observed peat depth of 2.06 m reported by Kokfelt et al. (2010).

The model initially had an uneven surface where the majority of the patches were suitable for moss growth because of the shallow peat depth and an annual WTP near the surface (Figs. 5e and 6a). Comment: Did this unevenness persist? The site is a palsa mire; did the model simulate cycles of palsa mound development and decay?

Response: The uneven surface persists (see Fig. 2 in this document) though heterogeneity increased and then decreased later to stabilize over time but we didn't notice palsa mound development because the ice expansion processes is not included in the model (an intended future modification).

We used these basal dates to start our model simulations. In the STD experiment, the simulated cumulative peat depth profile for the last 4700 years is consistent with the observed peat accumulation pattern (Kokfelt et al., 2010). The average increase in peat depth was simulated to be 2.11 m, which can be compared with the observed increase in peat depth of 2.06 m (Fig. 4). The simulated trajectory of the cumulative peat depth is also comparable to the observed data. In VLD ex Comments: Some repetition here of what is said in the previous section (see lines: 501-504 in the RM). Response: Thanks, we have now removed that part and revised the sentence. Revised

text: We used these basal dates to start our model simulations. In the STD experiment, the simulated cumulative peat depth profile for the last 4700 years is consistent with the observed peat accumulation pattern (Kokfelt et al., 2010). In VLD experiment, the average increase in peat depth was simulated to be 4.2 m, which can be compared to 5 m of observed peat depth (Frolking et al., 2010).

Mosses emerged as the dominant PFT at the beginning of the simulation, while 300-400 years after peat inception shrubs started establishing in the higher elevated patches as a result of a lowering of WTP (Figs. 5e and 6a).

Comments: What about palsa formation and collapse? Is this not an area where such processes occur. These processes don't seem to be represented in the model. Response: You are right these processes are not represented in the model and will be included in the future modifications.

NPP in the first half of the 21st century, resulting in accelerated peat accumulation, but that the increase in decomposition outpaces the increase in NPP by around 2040, resulting in the loss of a substantial amount of carbon by the end of the 21st century (Fig 9). Comment: Okay, but peatlands have formed extensively in the temperate and boreal zones and many of these peatlands have a substantial bryophyte component in their flora. So, why will warmed Arctic and sub-Arctic peatlands lose carbon? Is it not possible that new peatlands will also develop? Perhaps much depends on local hydrological conditions.

Response: Yes, this is a very good point, so we have revised the sentence (see lines: 608-614 in the RM). We have found the similar finding in our companion paper

Companion paper (lines 21-30)- A majority of modelled peatland sites in Scandinavia, Europe, Russia and Central and eastern Canada change from carbon sinks through the Holocene to potential carbon sources in the coming century. In contrast, the carbon sink capacity of modelled sites in Siberia, Far East Russia, Alaska and western and northern Canada was predicted to increase in the coming century. The greatest

changes were evident in eastern Siberia, northwest Canada and in Alaska, where peat production, from being hampered by permafrost and low productivity due the cold climate in these regions in the past, was simulated to increase greatly due to warming, wetter climate and greater CO2 levels by the year 2100. In contrast, our model predicts that sites that are expected to experience reduced precipitation rates and are currently permafrost free will lose more carbon in the future.

Revised Text: Higher temperatures will result in earlier snowmelt and a longer growing season (Euskirchen et al., 2006), promoting plant productivity. Our results for both a strong warming (RCP8.5) and low warming (RCP2.6) scenario indicate that the limited increase in decomposition due to soil warming will be more than compensated by the increase in NPP in the first half of the 21st century, resulting in accelerated peat accumulation. Decomposition was, however, simulated to increase after 2040 due to permafrost thawing and high temperature, resulting in the loss of comparatively higher amount of carbon by the end of the 21st century (Fig. 12).

Figure 1-

Comment: Surface runoff in this figure seems to include subsurface flow in the peat layers. Also, AWTP needs formal definition - the reader should not have to guess its meaning.

Response: We have now revised this figure (see Fig. 4 in the RM) and included those components.

Figure 4-

Comment: The light red shaded area shows the 95% confidence interval (CI) inferred from the simulation data. It would be useful to explain somewhere how the CIs were calculated.

Response: Here is the calculation. We have included this information in footnotes (see line: 674 in the RM) CI = $\mu \pm$ Z.95 SE

where $\mu$ is the mean peat depth across all the patches, SE is the standard error of the mean and Z.95 is the confidence coefficient from the means of a normal distribution required to contain 0.95 of the area.

Fig. 9 The changes in peat thickness under the 'all' scenario are actually quite small.

Response: Yes, the change in peat thickness under the all scenario is small we have revised this in the text (see lines: 22-24 and 608-614 in the RM).

Revised Text: We found that the Stordalen mire may be expected to sequester more carbon in the first half of the 21st century due to milder and wetter climate conditions, a longer growing season, and the CO2 fertilization effect, turning into a carbon source after mid-century because of higher decomposition rates in response to warming soils.

[revised manuscript text omitted]